# Stabilized ε-Fe$_2$C catalyst with Mn tuning to suppress C1 byproduct selectivity for high-temperature olefin synthesis

Fei Qian[1,2,3,5], Jiawei Bai[1,2,3,5], Yi Cai[1,2,3,5], Hui Yang[1,2,3], Xue-Min Cao[1,2,3], Xingchen Liu ●[1] ✉, Xing-Wu Liu ●[2] ✉, Yong Yang[1,2,3], Yong-Wang Li[1,2,3], Ding Ma ●[4] & Xiao-Dong Wen ●[1,2,3] ✉

Accurately controlling the product selectivity in syngas conversion, especially increasing the olefin selectivity while minimizing C1 byproducts, remains a significant challenge. Epsilon Fe$_2$C is deemed a promising candidate catalyst due to its inherently low CO$_2$ selectivity, but its use is hindered by its poor high-temperature stability. Herein, we report the successful synthesis of highly stable ε-Fe$_2$C through a N-induced strategy utilizing pyrolysis of Prussian blue analogs (PBAs). This catalyst, with precisely controlled Mn promoter, not only achieved an olefin selectivity of up to 70.2% but also minimized the selectivity of C1 byproducts to 19.0%, including 11.9% CO$_2$ and 7.1% CH$_4$. The superior performance of our ε-Fe$_2$C-xMn catalysts, particularly in minimizing CO$_2$ formation, is largely attributed to the interface of dispersed MnO cluster and ε-Fe$_2$C, which crucially limits CO to CO$_2$ conversion. Here, we enhance the carbon efficiency and economic viability of the olefin production process while maintaining high catalytic activity.

Among the products of Fischer-Tropsch synthesis (FTS), which is a critical process for converting syngas—sourced from coal, natural gas, and biomass—into liquid fuels and chemicals, olefins stand out due to their widespread use in the chemical industry[1–3]. Although nano prismatic Co$_2$C and Ru-Na catalysts have demonstrated high efficiency in olefin production, their industrial application remains constrained by their higher costs relative to iron and comparatively low hydrocarbon space-time yields[4,5]. In contrast, iron-based catalysts, owing to their low cost, operational flexibility, and tunable product distributions, are extensively employed in industrial FTS processes[6,7]. However, issues such as high CH$_4$ or CO$_2$ selectivity and limited olefin selectivity often compromise the carbon efficiency and economic viability of these catalysts. Therefore, there is a growing demand to develop efficient catalysts that can achieve high olefin selectivity while minimizing C1 byproduct formation[8,9].

To improve the efficacy of iron-based catalysts in olefin synthesis, extensive research has focused on exploring diverse catalyst formulations and optimizing the reaction conditions[10]. Notably, two prevalent and effective strategies in reported works include incorporation of an alkali (K, Na) and manganese (Mn) as promoters to inhibit hydrogenation and elevation of the reaction temperature to facilitate olefin desorption[11–13]. However, elevated reaction temperatures and the presence of alkali promoters, while beneficial for olefin selectivity, tend to enhance the water–gas shift (WGS) reaction, leading to an increase in CO$_2$ selectivity. For instance, in the case of Fe-ZnO catalysts modified with sodium (Na), up to 35% of CO was converted to CO$_2$[14]. At 613 K, the CO$_2$ selectivity reached 40% on iron catalysts regulated by Al$_2$O$_3$[10], and it increased to 52.0% on Fe-K catalysts supported by reduced graphene oxide (rGO)[15]. Moreover, although Mn has been extensively studied as a promoter to enhance olefin selectivity,

[1]State Key Laboratory of Coal Conversion, Institute of Coal Chemistry, Chinese Academy of Sciences, Taiyuan 030001, China. [2]National Energy Center for Coal to Liquids, Synfuels China Co., Ltd., Huairou District, Beijing 101400, China. [3]University of Chinese Academy of Sciences, No. 19A Yuquan Road, Beijing 100049, PR China. [4]Beijing National Laboratory for Molecular Sciences, New Cornerstone Science Laboratory, College of Chemistry and Molecular Engineering, Peking University, Beijing, China. [5]These authors contributed equally: Fei Qian, Jiawei Bai, Yi Cai. ✉e-mail: liuxingchen@sxicc.ac.cn; liuxingwu@sxicc.ac.cn; wxd@sxicc.ac.cn

according to our knowledge, existing literature on the Fe-Mn system indicates that while Mn promoters do increase olefin selectivity, they do not significantly reduce $CO_2$ selectivity[16–21].

$CO_2$ primarily originates from the undesired WGS side reaction. Therefore, one effective strategy to reduce $CO_2$ selectivity is to prevent water molecules produced during FTS from recontacting the catalyst surface. Recently, surface hydrophobicity modification of $\chi$-$Fe_5C_2$-Mn catalysts[22], as demonstrated by Ding et al., has proven effective in suppressing readsorption of $H_2O$, leading to a notable decrease in $CO_2$ selectivity to 15%. However, this approach entails a compromise: the hydrophobic $SiO_2$ layer, applied to the catalyst surface to prevent water recontact, inadvertently covers some active sites, thereby potentially diminishing the catalyst activity. An alternative strategy worth exploring is direct reduction or elimination of the active sites responsible for the WGS reaction on the catalyst, which is a potential method to enhance the carbon utilization efficiency while maintaining a high catalytic activity.

While there is a consensus that $Fe_3O_4$ is an active phase for the WGS reaction during FTS, numerous studies have also confirmed that iron carbides, especially $\chi$-$Fe_5C_2$, exhibit WGS activity[23,24]. This insight opens up a potential strategy to reduce $CO_2$ selectivity by preventing catalyst oxidation and selectively fostering the formation of iron carbide phases with lower WGS activity. Epsilon iron carbide ($\varepsilon$-$Fe_2C$ or $\varepsilon$-$Fe_{2.2}C$), which is a phase commonly observed in low-temperature Fe-based FTS processes, has emerged as a promising candidate[25,26]. Its inherently low $CO_2$ selectivity makes it an attractive option for selective olefin production[26–29]. Considerable research has been dedicated to maintaining the stability of $\varepsilon$-$Fe_2C$ by carefully controlling the pretreatment and carburization conditions. Xu et al. directly prepared $\varepsilon$-$Fe_2C$ at low temperatures using quenched skeletal iron as a precursor, confirming its excellent low-temperature FTS activity[27]. Fu et al. synthesized $Fe_2N/Al_2O_3$ and observed the transformation of $Fe_2N/Al_2O_3$ to $Fe_2C/Al_2O_3$ during the FTS reaction process[29]. However, maintaining the stability of $\varepsilon$-$Fe_2C$ under the high-temperature conditions encountered in FTS reactions has been a significant challenge. Recently, Wang et al. achieved stability of $\varepsilon$-$Fe_2C$@graphene catalysts under high-temperature FTS conditions by confining pure-phase $\varepsilon$-$Fe_2C$ nanocrystals within graphene layers[28]. Despite these advancements, there is still a risk of phase transition or even oxidation of $\varepsilon$-$Fe_2C$ not shielded by graphene. To date, there have been no reported efforts to achieve high olefin selectivity along with a low amount of C1 byproducts by using $\varepsilon$-$Fe_2C$ as the primary active phase under high-temperature FTS conditions.

Herein, we report the preparation of Mn-modified $\varepsilon$-$Fe_2C$ catalysts through the pyrolysis of PBAs, which outperform existing state-of-the-art $\varepsilon$-$Fe_2C$ systems by producing higher value-added olefins and suppressing C1 byproducts. A standout feature of our system is its olefin selectivity, which constitutes 70.2% of the total product yield. The $\varepsilon$-$Fe_2C$-Mn system not only showcases optimal olefin selectivity but also demonstrates remarkable efficiency in reducing C1 byproducts, with $CO_2$ selectivity at just 11.9% and overall C1 byproduct selectivity below 20%. This performance is notably superior to the 22% C1 byproduct selectivity reported by Ding et al. using a hydrophobic catalyst[22]. Our research has expanded the operational scope of single-phase $Fe_2C$, providing valuable insights into designing Fischer-Tropsch synthesis to olefins (FTO) catalyst with high carbon utilization efficiency.

## Results

### Structural characterizations

Figure 1 illustrates our proposed approach for preparing ultrastable $\varepsilon$-$Fe_2C$-based catalysts. Initially, $\varepsilon$-$Fe_2N$ was obtained by thermally decomposing Prussian blue precursors (Fig. 1a and d) at 550 °C in an ammonia atmosphere, followed by a carbonization phase transition to form $\varepsilon$-$Fe_2C$ under syngas conditions at 300 °C (Fig. 1c and f). In situ X-ray diffraction (XRD) was employed to monitor the pyrolysis process

within the ammonia atmosphere, capturing the conversion of $Fe_4[Fe(CN)_6]_3$ to $Fe_2[Fe(CN)_6]$ at approximately 300 °C and its subsequent transition to $\varepsilon$-$Fe_2N$ at approximately 400 °C (see Supplementary Fig. 1). Owing to their identical crystal structures, $\varepsilon$-$Fe_2N$ and $\varepsilon$-$Fe_2C$ are challenging to differentiate via XRD. However, their magnetic properties provide distinctive features—$\varepsilon$-$Fe_2N$ is nonmagnetic, while $\varepsilon$-$Fe_2C$ exhibits magnetism. Mössbauer spectroscopy (MES) provides data that enable straightforward differentiation, with $\varepsilon$-$Fe_2N$ corresponding to a singlet and $\varepsilon$-$Fe_2C$ to a sextet in the spectrum (illustrated in Fig. 1e and f). MES revealed a rapid transition from $\varepsilon$-$Fe_2N$ to $\varepsilon$-$Fe_2C$ within 2 hours under the reaction conditions (see Supplementary Fig. 2 and Table 1), supporting Fu et al.'s observation of $Fe_2C/Al_2O_3$ formation from $Fe_2N/Al_2O_3$ under FTS conditions[29].

To verify the enhanced thermal stability of $\varepsilon$-$Fe_2C$ synthesized via our method compared to that of $\varepsilon$-$Fe_2C$ obtained through direct carburization of $\alpha$-Fe, we conducted a comparative study using in situ XRD to examine their phase transitions under a programmed temperature increase in an inert atmosphere. Figure 2a, b display the in situ phase transition processes of $\varepsilon$-$Fe_2C$ synthesized from PBAs and $\alpha$-Fe, respectively. Notably, $\varepsilon$-$Fe_2C$ derived from the PBAs demonstrated significantly superior stability, maintaining its structure up to 440 °C. This level of stability, not previously reported in seminal research on $\varepsilon$-$Fe_2C$[26–29], marks a significant advancement. This stability at 440 °C ensures that the $\varepsilon$-$Fe_2C$ phase remains stable under typical FTS conditions. As the temperature increased, carbon depletion occurred, leading $\varepsilon$-$Fe_2C$ to transition into $\varepsilon$-$Fe_xC$ and subsequently into $\theta$-$Fe_3C$, with $\alpha$-Fe prevailing at higher temperatures. We directly synthesize $\varepsilon$-$Fe_2C$ via the carburization of $\alpha$-Fe at relatively low temperatures (180 – 200 °C), employing an $H_2/CO$ mixture. The synthesis approach and characteristics of our $\varepsilon$-$Fe_2C$ closely align with those reported by Xu et al.[27]. Therefore, we believe this type of $\varepsilon$-$Fe_2C$ adequately represents the typical properties of $\varepsilon$-$Fe_2C$ without N. As shown in Fig. 2b, $\varepsilon$-$Fe_2C$ produced from direct carburization of $\alpha$-Fe with $H_2/CO$ was unstable above 320 °C. Although Wang et al. suggested that encapsulating $\varepsilon$-$Fe_2C$ within graphene layers could anchor this metastable phase[28], high-resolution transmission electron microscopy (HRTEM) (Fig. 2c) did not confirm encapsulation of our synthesized $\varepsilon$-$Fe_2C$ by graphene, indicating that the observed stability enhancement of $\varepsilon$-$Fe_2C$ may not stem from surface graphene encapsulation. Given that our synthesis strategy involves the use of $\varepsilon$-$Fe_2N$ as a template, with simultaneous removal of N atoms and infusion of C atoms to yield $\varepsilon$-$Fe_2C$, we propose that N atoms significantly contribute to the thermal stability of $\varepsilon$-$Fe_2C$. Even though N atoms are swiftly eliminated during the early stages of the high-temperature carburization, some N atoms likely persist within the $\varepsilon$-$Fe_2C$ bulk. Since N (or C) occupies the octahedral sites of $\varepsilon$-$Fe_2N$ (or C) and given that N has a smaller atomic radius than C, the presence of N atoms alleviates the lattice stress induced by octahedral site occupancy, thereby bolstering the thermodynamic stability of $\varepsilon$-$Fe_2C$. Our experimental modification of the PBAs pyrolysis atmosphere validated this hypothesis. As depicted in Supplementary Figs. 3a, b and Table 2, $\varepsilon$-$Fe_2C$ could indeed be synthesized using our proposed the pyrolysis of PBAs method under both He and a 5% CO/ 95%He atmosphere. However, these atmospheres led to significantly reduced FTS reaction stability, accompanied by phase transformation of the catalysts (Supplementary Fig. 3c and Table 3). It is interesting to explore the underlying mechanisms of the N-induced stability enhancement of $\varepsilon$-$Fe_2C$. Through X-ray energy dispersive spectroscopy (EDS) image and XPS characterization results (Supplementary Fig. 4), we concluded that N atoms predominantly reside inside the catalysts, with no presence of N on the surface. Concurrently, EDS quantitative analysis confirmed the minimal content of N, with an N/Fe ratio of merely about 2.5% (Supplementary Table 4). To further elucidate the effect of N on catalyst stability, we have performed DFT calculations to assess its influence on the catalyst's formation energy. These calculations indicate that the incorporation of N

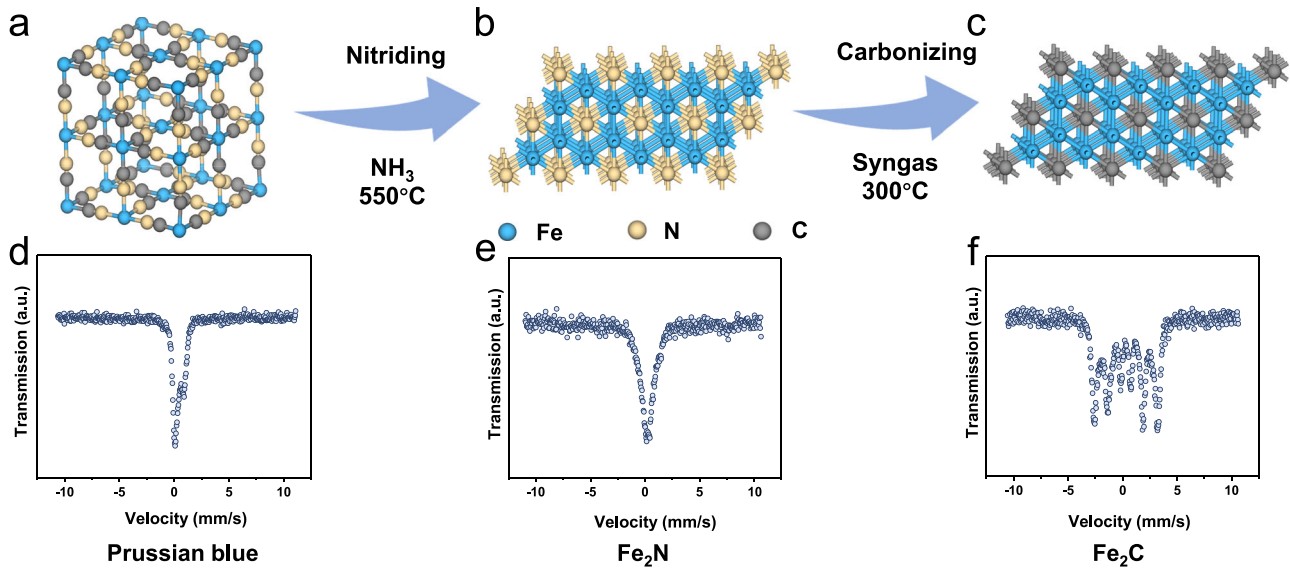

**Fig. 1 | Schematic illustration of the strategy used to synthesize ε-Fe₂C catalysts.** Models of **a** Prussian blue precursors, **b** activated Fe₂N and **c** carbonized Fe₂C. Mössbauer spectra of **d** Prussian blue precursors, **e** activated Fe₂N and **f** carbonized Fe₂C.

lowers the formation energy, thereby augmenting the catalyst's stability (Supplementary Fig. 5). Even under conditions of elevated temperature, pressure, and CO partial pressure, where the N concentration tends to diminish, it still markedly improves the stability of our catalyst.

To increase the selectivity for olefins among FTS products, the addition of promoters is a common strategy, with manganese (Mn) being an excellent choice. PBAs were synthesized through a coprecipitation method, during which various amounts of the Mn promoter were introduced. A range of catalysts with different Mn addition levels (ε-Fe₂N-xMn, where x = 0, 0.02, 0.04, 0.06, 0.2, 0.4, and 0.6) were prepared using this approach. The XRD patterns (Supplementary Fig. 6a) revealed the presence of only the ε-Fe₂N phase. Furthermore, with increasing Mn content, the (001) diffraction peak shifted to lower angles, indicating high dispersion of Mn and partial Mn doping into the ε-Fe₂N lattice. MES further confirmed that the presence of Mn did not affect the stability of ε-Fe₂N (refer to Supplementary Fig. 7 and Table 5).

Under Fischer–Tropsch reaction conditions (T = 300 °C, P = 2 MPa), the ε-Fe₂N-xMn catalysts underwent carbonization to ε-Fe₂C-xMn (see Supplementary Fig. 6b). The XRD patterns and MES spectra of the catalysts after FTS confirmed the emergence of ε-Fe₂C (JCPDS#36-1249) (Fig. 2e, f and Supplementary Table 6). Furthermore, the Mn promoter did not lead to the generation of any other iron oxides or carbides, indicating that ε-Fe₂C was the active phase for the Fischer-Tropsch reaction, while Fe₂N served as a precursor for high-stability ε-Fe₂C formation. According to Supplementary Fig. 6, no diffraction peaks corresponding to Mn species were observed at low Mn contents, indicating that Mn remained highly dispersed. However, at a Mn content of 0.2, MnO peaks (PDF#75-0257) emerged in the XRD pattern, suggesting that at elevated Mn loadings, Mn may exist in two forms: highly dispersed on the catalyst surface and phase-separated as a nucleated MnO phase.

X-ray absorption spectroscopy (XAS) and X-ray photoelectron spectroscopy (XPS) were employed for a more in-depth investigation of the chemical compositions and electron states of the ε-Fe₂C-xMn catalysts. Highly analogous FT EXAFS spectra were obtained, suggesting that the local geometric structure of the Fe species was unchanged, as shown in Fig. 2h. The changes in the electronic structure were further analyzed via XANES spectroscopy, as shown in Fig. 2g. The absorption edge distinctly shifted toward a lower energy level,

approaching that of Fe foil, for both the Mn-promoted and unpromoted samples, suggesting a reduction in the oxidation state of the Fe atoms. To further investigate the catalyst surface compositions, XPS was employed. The Fe 2p spectra for all the samples are shown in Supplementary Fig. 8, with peaks at 706.3 and 710.5 eV corresponding to Fe⁰ and Fe²⁺/Fe³⁺, respectively. The addition of manganese led to a significant reduction in Fe²⁺/Fe³⁺ species in both the fresh and spent catalysts. This result indicates that manganese acts as an inhibitor, impeding the transformation of iron carbides into iron oxides on the catalyst surface. Furthermore, as shown in the TEM images (Fig. 2d), ε-Fe₂C-0.4Mn was uniformly dispersed, with an average crystallite size of approximately 57.4 nm. The particle size distribution did not change with the addition of Mn (Supplementary Fig. 9). HRTEM images (Fig. 2c) confirmed the presence of ε-Fe₂C with a (010) interplanar distance of approximately 2.40 Å after the reaction. At the same time, the introduction of the Mn promoter did not alter the stability of the ε-Fe₂C phase. The ε-Fe₂C phase remains stable at the same temperature, regardless of whether the manganese promoter is introduced or not (Supplementary Fig. 10).

## Catalytic performance

The catalysts were compared under the same CO conversion conditions (41.5 ± 1.5%) at 573 K, 2.0 MPa, H₂/CO = 2.5, and GHSV varied from 20 to 500 L·g$_{cat}$⁻¹·h⁻¹. With increasing Mn loading, the selectivity for C1 byproducts (CH₄ + CO₂) tended to decrease from 66.7% to 19.0% (Fig. 3a and Supplementary Table 7). Moreover, the Mn promoter significantly enhanced the selectivity for C₂₊ olefins. When the Mn/Fe molar ratio reached 0.2:1 (ε-Fe₂C-0.4Mn), the selectivity for C1 products reached the lowest levels, with CO₂ at 11.9% and CH₄ at 7.1%, while C₂₋₄ o/p reached 7.2, and C₂₊ olefins accounted for 70.2% of the total products. Moreover, the conversion of CO decreased from 98.1% to 36.1% with increasing Mn loading when the catalysts were investigated at the same GHSV of 20 L·g$_{cat}$⁻¹·h⁻¹ (Fig. 3a). In addition to the Mn/Fe molar ratio, the H₂/CO ratio also affected the product distribution. A higher H₂/CO ratio was beneficial for CO conversion, reducing the production of CO₂ and generally increasing the generation of CH₄ (Supplementary Fig. 11a, Supplementary Table 8). Increasing the reaction pressure promoted CO conversion, which increased to 42% at a pressure of 2.5 MPa (Supplementary Fig. 11b, Supplementary Table 9). Furthermore, temperature was also a significant factor in the reaction conditions. With an increase in temperature, the CO

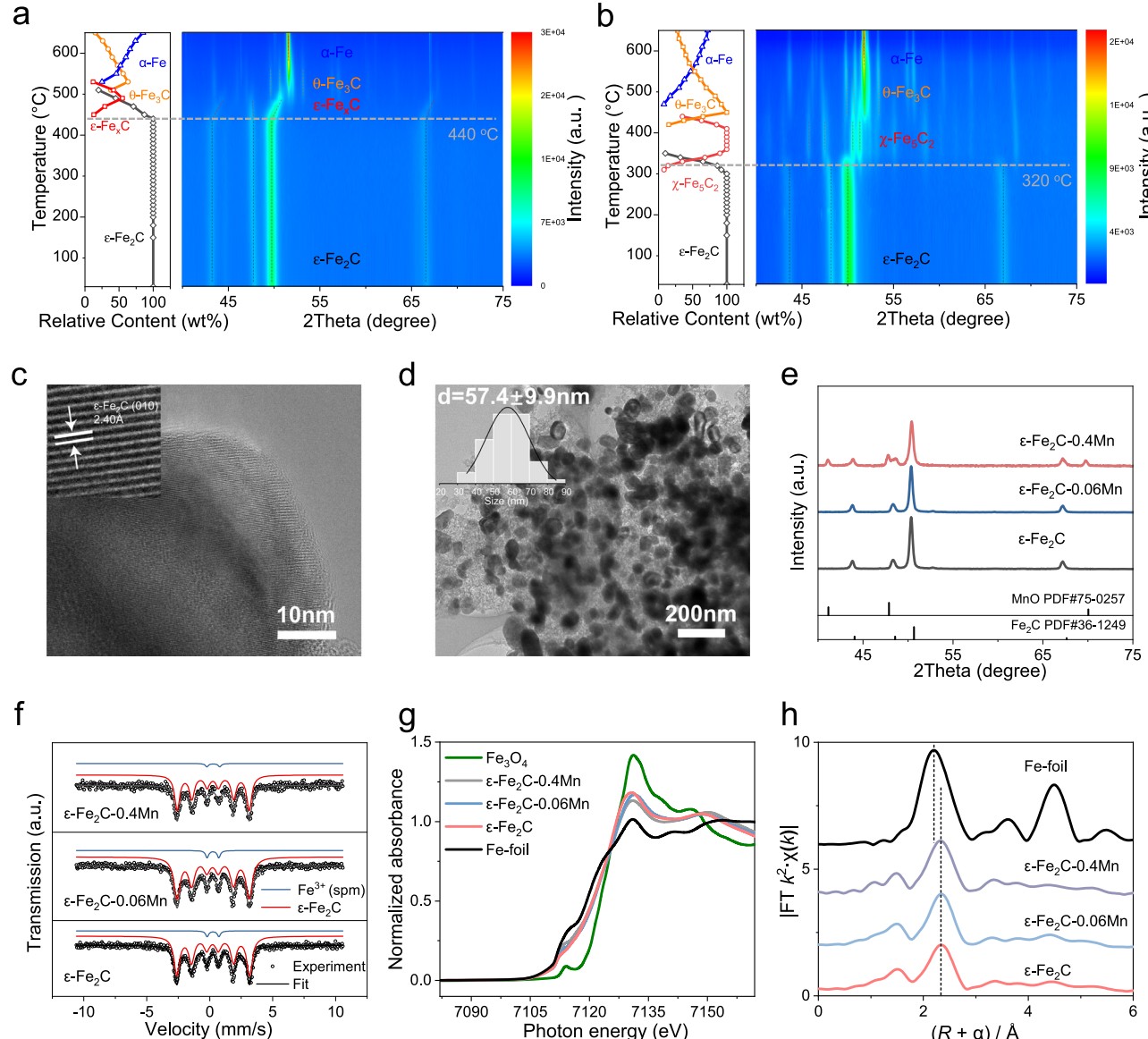

**Fig. 2 | Comprehensive characterization of spent ε-Fe₂C-xMn catalysts.** In situ XRD patterns of the temperature-programmed phase transition of two ε-Fe₂C samples in a helium atmosphere from room temperature to 650 °C: **a** ε-Fe₂C derived from PBAs using the method proposed in this work and **b** ε-Fe₂C derived from α-Fe fully carburized with H₂/CO. **c** HRTEM image of the ε-Fe₂C-0.4Mn catalyst. **d** TEM image and particle size distribution for the ε-Fe₂C-0.4Mn catalyst. **e** XRD patterns, **f** Mössbauer spectra, **g** iron K-edge X-ray absorption near-edge structure (XANES) spectra, and **h** Fourier transform extended X-ray absorption fine structure (FT EXAFS) at the Fe K-edge for ε-Fe₂C-xMn catalysts (where x = 0, 0.06, and 0.4).

conversion and production of C1 products increased (Supplementary Fig. 11c, Supplementary Table 10). Moreover, a comparative analysis with extant literature on Mn promoters (Supplementary Table 11) reveals that most studies report a selectivity for C1 by-products exceeding 35%. Our results not only align with but also significantly advance past findings on Mn modulated Fe-based FTS catalysts by showing that our Fe₂C-Mn system achieves a CO₂ selectivity of only 11.9% and a total C1 by-product selectivity of less than 20%. To our knowledge, existing literature on the Fe-Mn system consistently reports that Mn promoters enhance olefin selectivity, but there are no reports of a significant reduction in CO₂ selectivity.

We also compared our results with the state-of-art syngas-to-olefin process and the performance of other previous seminal single phase ε-Fe₂C. (Fig. 3b, Supplementary Fig. 12) Our ε-Fe₂C-Mn catalysts have demonstrated superior performance compared to existing state-of-the-art ε-Fe₂C systems. A key feature of our system is its olefin selectivity, which accounts for 70.2% of the total product yield.

(Supplementary Fig. 12). Moreover, our ε-Fe₂C-Mn system excels in reducing C1 byproducts, with CO₂ selectivity at a mere 11.9% and overall C1 byproduct selectivity about 19%. (Supplementary Table 12). This performance is notably superior to the 22% C1 byproduct selectivity reported by Ding et al.[22] using a hydrophobic catalyst (Fig. 3b). The ε-Fe₂C reported by Wang et al.[26] shows a CO₂ selectivity of 5% and a methane selectivity of 17% under low-temperature conditions (235 °C and 15% CO conversion). In contrast, even at a higher temperature of 280 °C, our ε-Fe₂C-Mn catalyst demonstrates only 2.4% CO₂ selectivity and 9.9% methane selectivity. Furthermore, the stability of the catalysts was evaluated, as shown in Fig. 3c. The catalytic performance of ε-Fe₂C-0.4Mn was tested under the following reaction conditions: 573 K, 2.0 MPa, H₂/CO = 2.5, and GHSV = 20 L·gcat⁻¹·h⁻¹. The catalysts exhibited stable activity and product distribution over 120 hours of continuous operation. As depicted in Supplementary Fig. 13, the product distribution of the ε-Fe₂C-0.4Mn catalyst is dominated by alkenes. The products are primarily distributed below C12. This distribution pattern

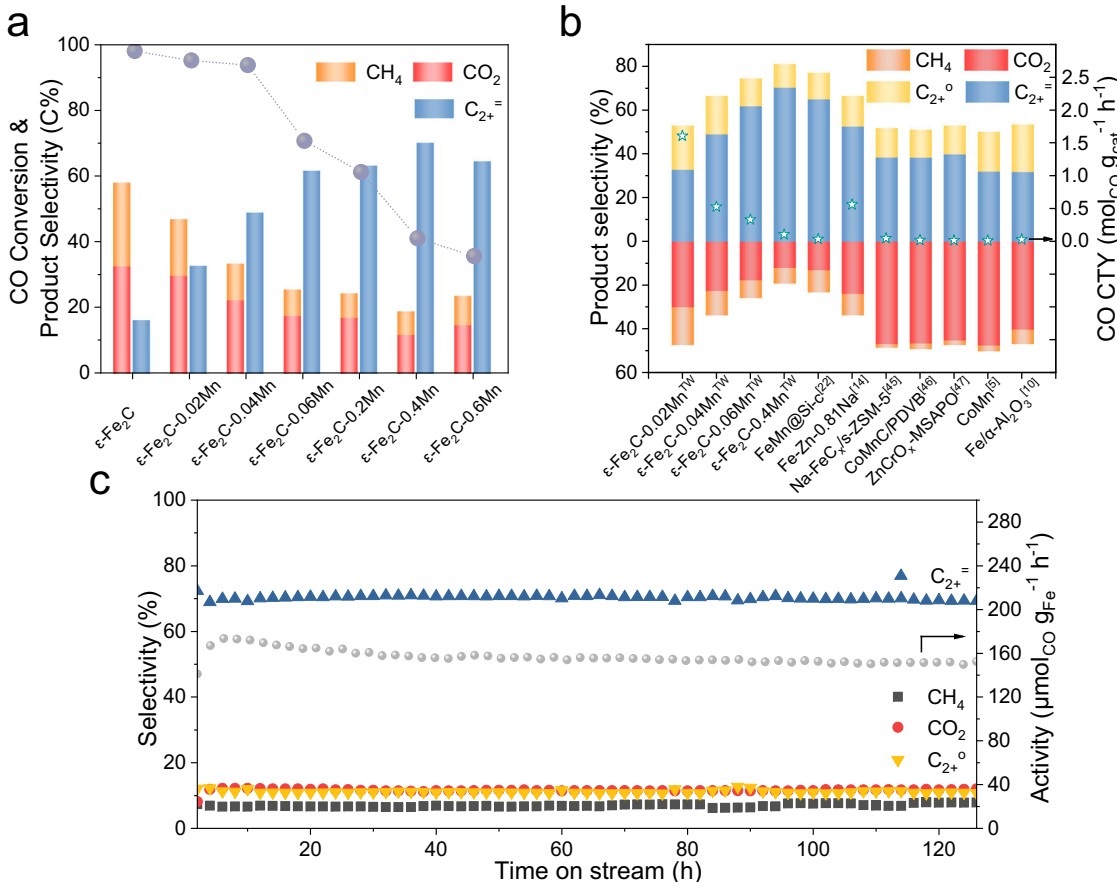

**Fig. 3 | Catalytic performance. a** CO conversion and product selectivity for ε-Fe₂C with various manganese addition levels. (Reaction conditions: 0.10 g of catalyst, 573 K, 2.0 MPa, H₂/CO = 2.5, and gas hourly space velocity (GHSV) varied from 20 to 500 L·g$_{cat}^{-1}$·h$^{-1}$ for a CO conversion rate of 41.5 ± 1.5%.) **b** Comparison of the catalytic performance of ε-Fe₂C-xMn with that of other previously reported catalysts[5,10,14,22,45-47]. **c** Stability test for the ε-Fe₂C-0.4Mn catalyst (reaction conditions: 0.10 g of catalyst, 573 K, 2.0 MPa, H₂/CO = 2.5, GHSV = 20 L·g$_{cat}^{-1}$·h$^{-1}$).

suggests that the ε-Fe₂C-Mn catalyst is highly suitable for the production of high-valued olefins. Overall, we report the preparation of Mn-modified ε-Fe₂C catalysts through the pyrolysis of PBAs, which are utilized for high olefin production from syngas while simultaneously suppressing C1 byproducts. The superior performance of our ε-Fe₂C-xMn catalysts, particularly in minimizing CO₂ formation, is largely attributed to the interface of the dispersed MnO cluster and the ε-Fe₂C phase.

### Structural information of the manganese promoter

To gain more comprehensive insights into the performance of manganese-promoted ε-Fe₂C catalysts, investigation of the structure of manganese within the catalyst is crucial. Analysis of the catalysts using inductively coupled plasma (ICP) and XPS data revealed that the Mn/Fe ratio on the catalyst surface is higher than that in the bulk. Interestingly, as the Mn loading increased, the Mn/Fe ratio on the catalyst surface after the reaction exceeded that of the fresh catalyst, suggesting the aggregation of manganese on the surface (Fig. 4a). Furthermore, XAS was conducted to elucidate the manganese structure (Fig. 4b and c). The normalized Mn K-edge XANES data indicated that the manganese in the catalysts with different Mn/Fe ratios was closer to MnO species. Additionally, the coordination number indicates that these MnO entities are not isolated atoms but rather clusters (MnO)ₙ, emphasizing their clustered structure (Supplementary Fig. 14 and Supplementary Table 13). XPS analysis of the valence state of manganese confirmed the presence of Mn²⁺ in both the fresh and spent ε-Fe₂C samples with various manganese additions (Fig. 4d), which was consistent with the XANES results (Fig. 4b). STEM images of the ε-Fe₂C

catalysts (Fig. 4e and f) revealed that in the ε-Fe₂C-0.06Mn sample, manganese dispersed uniformly, with only a small amount of manganese aggregating. In contrast, in the ε-Fe₂C-0.4Mn sample, although manganese was evenly dispersed, larger manganese particles were observed. The XRD patterns of the catalysts confirmed the presence of the MnO phase with increasing manganese content, further confirming the aggregation phenomenon of MnO. Upon analyzing the structure of the manganese additive, we observed a noticeable MnO aggregation phenomenon, particularly at higher manganese levels, resulting in the presence of distinct MnO phases. Further investigation was required to determine whether these separated MnO phases played a role in the reaction process. We conducted experiments by mechanically mixing fresh catalysts with commercial MnO, both before pelleting and after pelleting. The resulting catalysts were evaluated under identical reaction conditions. Surprisingly, the presence of separated MnO did not appear to impact the performance of ε-Fe₂C, as indicated by the evaluation results (see Supplementary Table 14). Therefore, the contributing factor to product selectivity appears to be the interface of highly dispersed MnO species with ε-Fe₂C rather than the aggregated MnO phase.

### Mechanism of the Mn promoter

The ε-Fe₂C-Mn catalysts exhibited notable suppression of C1 products, particularly CO₂. The generation of CO₂ has been widely investigated for iron-based FTS catalysts[13,26,30]. CO₂ generation is generally accepted to consist of primary CO₂ (CO* + O* → CO₂) and secondary CO₂ (CO + H₂O → CO₂ + H₂, WGS) pathways. The relative contribution of each pathway depends on various factors, such as the crystalline

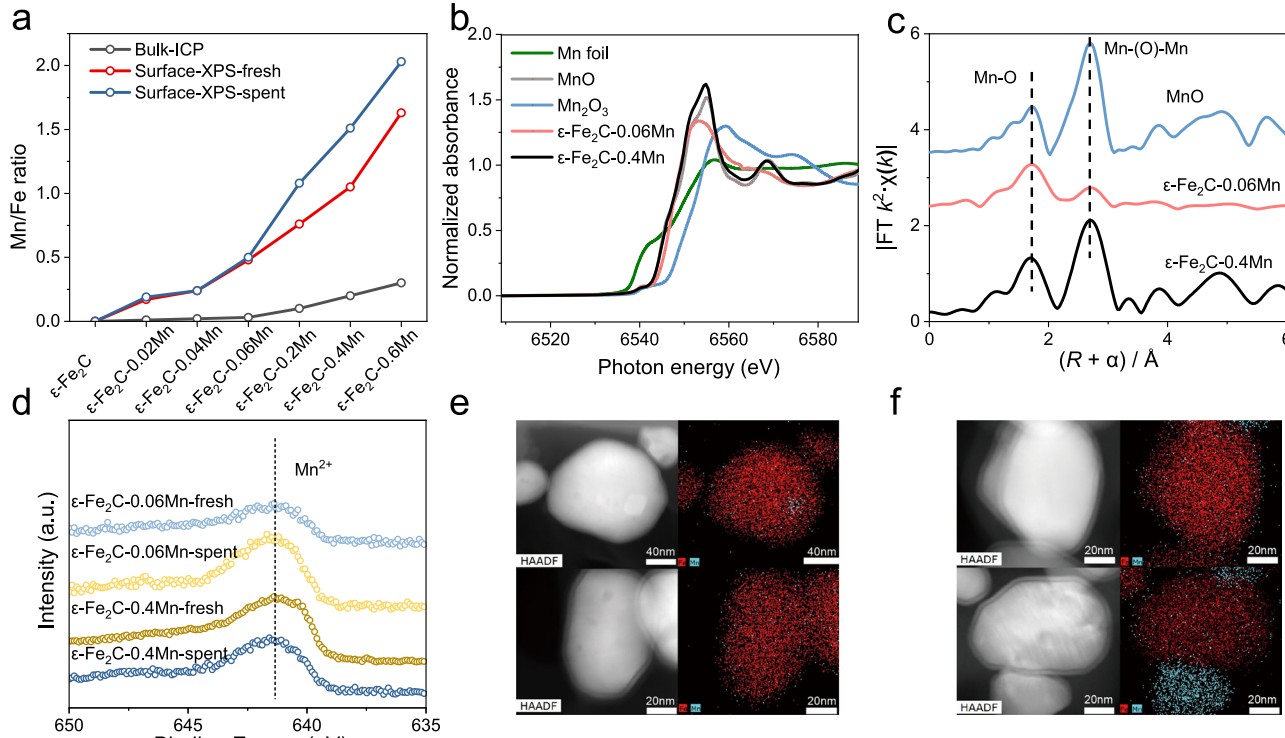

**Fig. 4 | Structural characterization of the manganese promoter. a** Surface and bulk Mn/Fe ratio of catalysts with various levels of manganese. Manganese K-edge XANES spectra **b** and EXAFS spectra **c** of ε-Fe$_2$C-xMn catalysts, with Mn$_2$O$_3$, MnO and Mn foil as references. **d** Mn 2$p$ XPS spectra of ε-Fe$_2$C-xMn catalysts. **e** HAADF and STEM-EDS elemental mapping images of ε-Fe$_2$C-0.06Mn after the FTS reaction. **f** HAADF and STEM-EDS elemental mapping images of ε-Fe$_2$C-0.4Mn after the FTS reaction.

phase, exposed surfaces, and pressure[30,31]. The CO$_2$ selectivity attributed to the primary pathway can be determined by extrapolating the CO$_2$ selectivity to zero CO conversion. As illustrated in Fig. 5a, the primary CO$_2$ pathway accounts for 5% ~ 6% of the CO$_2$ selectivity for the pure ε-Fe$_2$C phase. In contrast, the contribution of the primary CO$_2$ pathway for the ε-Fe$_2$C-0.4Mn catalyst drastically decreases to nearly zero, suggesting that the addition of manganese significantly inhibits the primary CO$_2$ formation process. To further explore the impact of manganese addition on secondary CO$_2$ formation, we carried out WGS pulse reactions. The addition of manganese resulted in a marked decrease in the WGS reaction (Fig. 5b), and the CO$_2$/CO intensity in the mass spectrum decreased from 0.085 to 0.03, implying that the addition of manganese also concurrently suppresses secondary CO$_2$ formation. Moreover, the temperature-programmed surface reaction confirmed the inhibitory effect of manganese addition in the WGS reaction (Supplementary Fig. 15). With increasing temperature, the ion currents of CO$_2$ and H$_2$ significantly decreased with the addition of manganese.

We implemented ab initio molecular dynamics (AIMD) simulations to gain a deeper understanding of the atomic-level mechanism behind the suppressed CO$_2$ selectivity caused by the highly dispersed manganese particles on ε-Fe$_2$C. Given the multiple potential active sites and reaction pathways on the supported catalyst, we utilized a modified ab initio nanoreactor approach[32,33] to initially estimate the reaction channels. The simulations commenced with a uniform mixture of CO and H$_2$ above the catalyst surfaces, with periodic forcing introduced to boost the frequency of collisions and barrier crossing. On a bare ε-Fe$_2$C (001) surface, many of the CO and H$_2$ molecules dissociated into adsorbed C*, O*, H*, and CH* (H-assisted dissociation), subsequently forming a range of surface species, including CH$_2$*, CH$_3$*, CH$_4$*, CHCO*, CCO*, and CO$_2$*, among others (Fig. 5c). The generation of CO$_2$ along the trajectories typically necessitates the combination of

a CO molecule with a surface O* atom. However, on a MnO/ε-Fe$_2$C (001) supported surface, CO$_2$ formation was not observed during the same simulation period (Fig. 5d). The conversion of surface species determined from the ab initio nanoreactor simulations is summarized in a simplified reaction network (Fig. 5e). We observed that incoming CO molecules tend to migrate and become trapped near the supported MnO clusters, thus impeding their reaction with surface O*.

These insights were corroborated by additional static DFT calculations, which revealed that the adsorption energy of CO on the MnO/ε-Fe$_2$C (001) interface was 0.43 eV greater than that on the bare surface (Supplementary Fig. 16). Furthermore, the barrier of the reaction (CO* + O* → CO$_2$*) significantly increased from 1.21 eV on the bare ε-Fe$_2$C (001) surface to 2.25 eV on the MnO/ε-Fe$_2$C (001) model. As an alternative CO$_2$ formation pathway, the reaction barrier for the carbonate mechanism (CO* + OH* → COOH*) on ε-Fe$_2$C (001)-MnO surface is 2.41 eV, which is significantly higher than the 1.15 eV observed on ε-Fe$_2$C (001) surface (Fig. 5f). The dissociation of H$_2$O into OH and finally O species serves as another source of surface O, which may contribute to CO$_2$ selectivity through secondary CO$_2$ formation. We thus also conducted calculations to assess the impact of manganese on the dissociation of H$_2$O into OH and O species on different models. As shown in Fig. 5f, the barrier to OH and O formation through H$_2$O dissociation is hardly affected by manganese (Supplementary Fig. 17). Therefore, the suppression of secondary CO$_2$ formation induced by manganese addition is probably also due to the elevated association barrier of CO with surface O and OH species.

To further understand the effect of the ε-Fe$_2$C-MnO interface on the olefin selectivity, we conducted pulse reactions to analyze the effect of the addition of manganese to the catalysts. The secondary hydrogenation ability of alkenes at the active site of the catalyst is a key factor affecting the olefin selectivity. An ethene pulse transient hydrogenation experiment was conducted to explore whether the

 

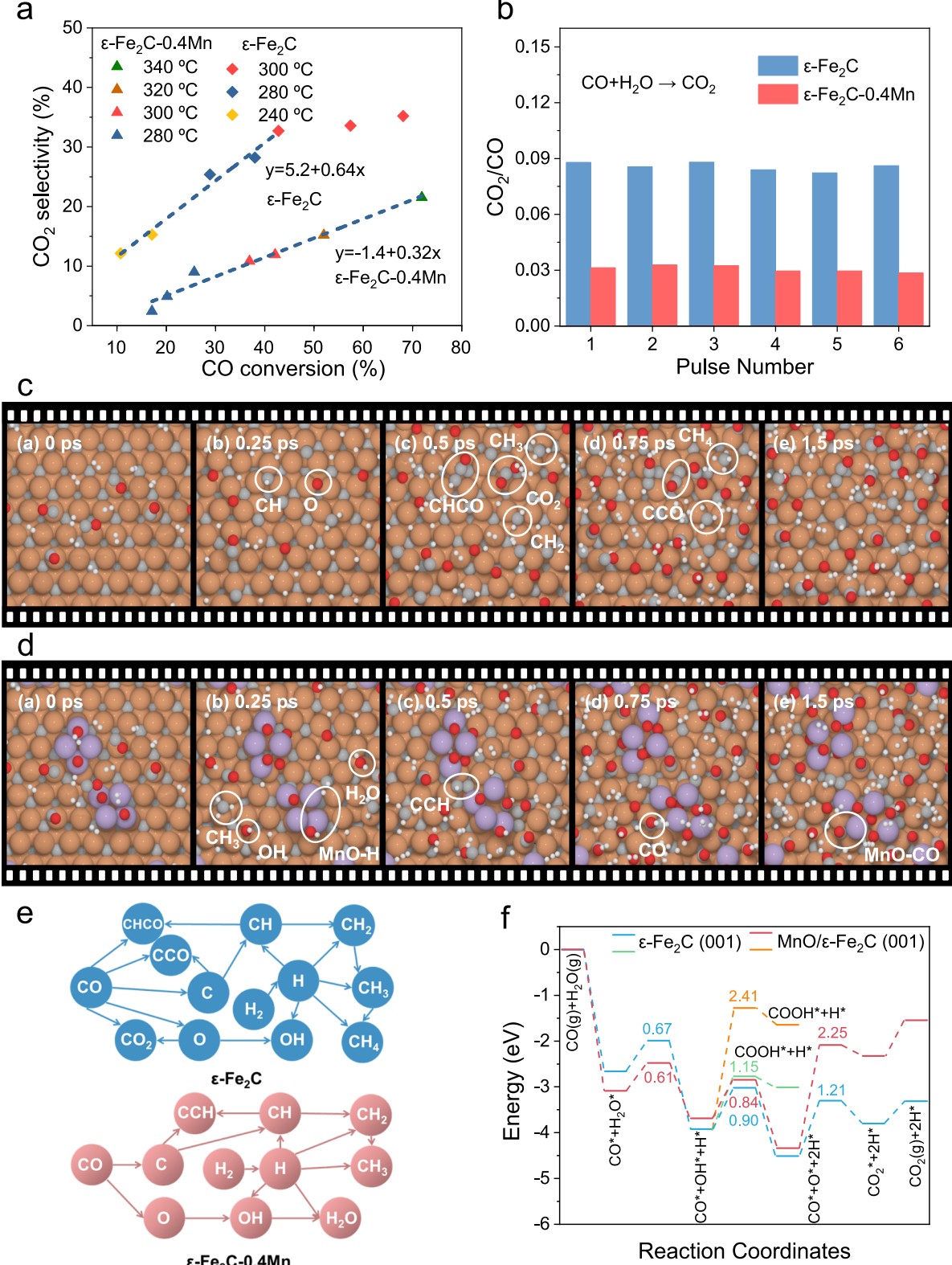

**Fig. 5 | Mechanism of the Mn promoter. a** $CO_2$ selectivity as a function of CO conversion. **b** Effect of Mn on the WGS reaction. **c** Typical nanoreactor trajectory on the ε-$Fe_2$C (001) surface: the simulation began with a collection of CO and $H_2$ above the surface (Fe: brown; C: gray; O: red; Mn: purple; H: white). The key surface species are circled to indicate the observed reactivity. **d** Typical nanoreactor trajectory on the MnO/ε-$Fe_2$C (001) surface. **e** The reaction network extracted from ab initio nanoreactor simulations. **f** The potential energy surface of $H_2O$ dissociation, OH dissociation, CO + O and CO + OH.

olefins preferentially desorbed from or hydrogenated on the ε-Fe$_2$C catalysts in the presence or absence of the manganese promoter. The catalysts were activated and reacted for 50 h under the reaction conditions, followed by switching to a H$_2$ flow at 573 K. Ethene was then pulsed into the system, and the products were detected by mass spectrometry. The R value represents the ratio of the integral area of alkenes to that of alkanes, indicating the ability of olefins to be hydrogenated. As shown in Supplementary Fig. 18c, the R value of ε-Fe$_2$C in the C$_2$H$_4$ hydrogenation reaction was 1.0, whereas the R value of ε-Fe$_2$C-0.4Mn was 2.0 (Supplementary Fig. 18d). Manganese was thus suggested to suppress the secondary hydrogenation of olefins. The same phenomenon was observed for the C$_3$H$_6$ hydrogenation reaction.

To gain further insights and corroborate the experimental findings, we examined the influence of manganese on the C$_2$H$_4$ hydrogenation reaction. Our ab initio nanoreactor trajectories revealed trapping of surface H* species on the MnO cluster, and the number of H* species adsorbed on the bare ε-Fe$_2$C (001) surface sites substantially decreased. This led to a diminished H* coverage on the surface, a finding supported by static calculations that demonstrated a reduction in the adsorption energy of H* on the surface with manganese addition (Supplementary Figs. 16 and 19). Furthermore, as suggested by the probe molecule (C$_2$H$_4$), the adsorption energy of olefins also significantly decreased from −1.50 eV to −0.67 eV with manganese addition, which greatly facilitates their desorption from the surface and helps avoid their secondary hydrogenation to paraffins through re-adsorption. Furthermore, additional barrier calculations of C$_2$H$_4$ hydrogenation indicate that the formation of C$_2$H$_6$ is kinetically suppressed on ε-Fe$_2$C (001)-MnO compared to ε-Fe$_2$C (001) without Mn (2.38 eV vs. 1.64 eV), significantly enhancing the selectivity towards ethylene (Supplementary Fig. 20). Both of these factors could potentially impede the further hydrogenation of olefins to paraffins, contributing to increased olefin selectivity.

## Discussion

In conclusion, we have introduced a groundbreaking advancement in the field of syngas conversion to olefins by successfully synthesizing a highly stable ε-Fe$_2$C catalyst via a nitrogen-induced strategy, utilizing PBAs as precursors. By incorporating a precisely controlled amount of Mn as a promoter, we have achieved a remarkable catalytic performance, marking a significant leap forward in the sustainable production of olefins. Our findings reveal that the synergy between the ε-Fe$_2$C structure and Mn promotion not only enhances olefin selectivity but also plays a critical role in suppressing the activity of WGS reaction, thereby reducing the formation of undesirable byproducts.

The development of the ε-Fe$_2$C catalyst represents a significant stride towards optimizing carbon efficiency and the economic viability of the olefin production process. By addressing the longstanding challenge of achieving high olefin selectivity alongside reduced C1 byproduct formation, this study paves the way for a more sustainable and efficient approach to olefin synthesis. Future research could explore the potential scalability of this catalyst system and investigate the integration of additional promoters or support materials to further enhance selectivity and stability for industrial applications.

## Methods
### Catalyst preparation

All the Fe-Mn catalysts with different Mn contents were synthesized by the coprecipitation method with Fe(NO$_3$)$_3$·9H$_2$O and Mn(NO$_3$)$_2$·4H$_2$O as the Fe and Mn sources, respectively. (NH$_4$)$_4$Fe(CN)$_6$ was used as the precipitating agent. Typically, a suitable amount of polyvinylpyrrolidone K30 (average molecular weight 40,000; TCI (Shanghai) Development Co., Ltd.) was dissolved in deionized water (200 g), and then, (NH$_4$)$_4$Fe(CN)$_6$ (Honeywell Specialty Chemicals Seelze GmbH) was added to the above solution, which was defined as solution

A. Then, different qualities of Fe(NO$_3$)$_3$·9H$_2$O and Mn(NO$_3$)$_2$·4H$_2$O (AR, Sinopharm Chemical Reagent Co., Ltd.) were dissolved in deionized water (200 g), which was defined as solution B. Nitrate solution B was slowly added to solution A under vigorous and continuous stirring to obtain different catalysts with Fe/Mn molar ratios of 100/0, 100/1, 100/2, 100/3, 100/10, 100/20 and 100/30, which were named herein as ε-Fe$_2$C, ε-Fe$_2$C-0.02Mn, ε-Fe$_2$C-0.04Mn, ε-Fe$_2$C-0.06Mn, ε-Fe$_2$C-0.2Mn, ε-Fe$_2$C-0.4Mn, and ε-Fe$_2$C-0.6Mn, respectively. The Fe/Mn ratio in various samples was measured by inductively coupled plasma-optical emission spectrometry (ICP-OES, Perkin Elmer).

### Catalyst characterization

A D8 powder diffractometer (Bruker, Karlsruhe, Germany) was applied to obtain XRD patterns using Co Kα radiation (λ = 0.179 nm) at 35 kV and 40 mA. A continuous mode was used with a scan step of 0.04° and a retention time of 0.4 s in the range of 40–75° for the catalysts.

In situ XRD experiments were carried out in an XRK900 reactor chamber (Anton-Paar Corporation) equipped on the diffractometer. Pyrolysis of Prussian blue was conducted in an ammonia atmosphere. The ε-Fe$_2$C obtained from Prussian blue was nitrided in a tube furnace and then carburized in a fixed-bed reactor. The ε-Fe$_2$C from α-Fe was directly carburized using H$_2$/CO (4:1) at a temperature of 180 °C for 48 hours in a tube furnace. The as-prepared Prussian blue and ε-Fe$_2$C solid powders were pressed into pellets and then packed into quartz capillaries. Pure NH$_3$ or He was fed at a flow rate of 30 ml min$^{-1}$, and the patterns were collected in continuous mode with a scan step of 0.04° and a retention time of 0.4 s in the range of 40–75° for the catalysts. The quantitative analysis of the XRD data was based on the reference intensity ratio (RIR) method using TOPAS software.

TEM was conducted on an FEI Talos F200A electron microscope at 200 kV. The samples were dispersed in ethanol by sonication and deposited on a copper grid with a porous carbon film, after which the sample was irradiated with an infrared lamp for 15 min before testing. High-angle annular dark-field scanning TEM (HAADF-STEM) was used to acquire STEM images (2048 × 2048 px) by using a camera length of 260 mm and a spot diameter of 0.5 nm.

XPS spectra were acquired with a Thermo Scientific K-alpha equipped with Al Kα (hν = 1486.6 eV) as the X-ray radiation source. The samples were prepared in a glove box to avoid oxidation. The C 1$s$ peak of carbon (284.6 eV) was used as a reference to calibrate the spectrum.

C$_3$H$_6$ and C$_2$H$_4$ pulse transient hydrogenation (PTH) experiments were performed with an AMI-300 equipped with a mass spectrometer. The catalysts were activated under ammonia gas, followed by switching to 10% H$_2$/Ar (50 mL/min), after which the temperature was changed to 300 °C. Subsequently, C$_3$H$_6$ or C$_2$H$_4$ was pulsed into the system to complete PTH. The effluent was monitored for C$_3$H$_6$ (m/z = 42) and C$_3$H$_8$ (m/z = 44) or for C$_2$H$_4$ (m/z = 26) and C$_2$H$_6$ (m/z = 30) using a PFEIFFER Omnistar mass spectrometer. The integrated peak area ratio of C$_3$H$_6$/C$_3$H$_8$ detected by the mass spectrometer was calculated and denoted as R, which represents the hydrogenation capacity of different catalysts.

WGS pulse reactions were performed with an AMI-300 equipped with a mass spectrometer. The catalysts were activated under ammonia gas, followed by switching to He (50 mL/min), after which the temperature was changed to 300 °C. Then, 5% CO/He and a bubbling bottle were used to perform the WGS pulse reaction. CO (m/z = 28) and CO$_2$ (m/z = 44) were monitored using a PFEIFFER Omnistar mass spectrometer.

The WGS temperature-programmed surface reaction (WGS-TPSR) was performed with an AMI-300 equipped with a mass spectrometer. The catalysts were activated under ammonia gas, followed by switching to 5% CO/He bubbling water (50 mL/min). Then, the temperature was raised to 400 °C. H$_2$ (m/z = 2) and CO$_2$ (m/z = 44) were monitored using a PFEIFFER Omnistar mass spectrometer.

X-ray absorption fine structure (XAFS) data were obtained at the BL14W1 of the Shanghai Synchrotron Radiation Facility (SSRF), China, which was operated at 3.5 GeV with a maximum current of 260 mA. The energy was calibrated to the absorption edge of pure Fe foil, and the fluorescence mode was used to obtain XAFS data.

Mössbauer experiments were conducted using an MR-351 constant acceleration transmission spectrometer at 10 K with 25 mCi $^{57}$Co in a Rh matrix. The phase composition was identified based on the isomer shift (IS), quadruple splitting (QS) and magnetic hyperfine field ($H_{hf}$) parameters. The content of each phase was determined from the adsorption peak areas based on the assumption of the same recoil-free factor for all kinds of iron nuclei in the catalyst.

## Catalyst testing

The catalytic performance of the different catalysts was evaluated in a fixed-bed microreactor with four stainless-steel tubes. The catalyst (100 mg) was diluted with quartz sand (200 mg) and was loaded into a 10 mm inner diameter quartz tube with an inserted stainless-steel sleeve to monitor the reaction temperature. The reaction conditions were 573 K, 2.0 MPa, $H_2/CO = 2.5$, and 20000 mL·$g_{cat}^{-1}$·$h^{-1}$ unless otherwise specified. After passing through a hot trap (433 K) and a cold trap (273 K), the tail gas was detected online by an Agilent 7890B chromatograph (GC). $C_1$-$C_4$ hydrocarbons were analyzed with two Gaspro capillary columns connected to a flame ionization detector (FID). $C_{4-7}$ hydrocarbons were analyzed with a PONA capillary column connected to an FID. The gases $CO_2$, Ar, and CO were analyzed through the coordination of a PLOT/Q, a 5 A molecular sieve and a Haysep Q capillary column connected to a thermal conductivity detector (TCD). The aqueous products, liquid oil products, and solid wax products were collected from the cold trap and hot trap and subsequently analyzed offline with an HP-PONA 19091s-001 column. The conversion and selectivity on the ε-$Fe_2C$ catalysts were calculated on a carbon basis. The carbon balance was in the range of 100 ± 5%.

The CO conversion ($X_{CO}$), product selectivity ($S_i$), reaction rate (R) and $C_{2+}$ olefin selectivity ($S_{C_{2+}^=}$) were calculated by the following equations:

$$X_{CO} = \frac{CO_{inlet} - CO_{outlet}}{CO_{inlet}} \times 100\% \tag{1}$$

$$S_i = \frac{N_i \times n_i}{\sum (N_i \times n_i)} \times 100\% \tag{2}$$

$$R_{CO} = \frac{WHSV \times X_{CO} \times C_{co}}{22400 \times \xi(Fe)} \times 100\% \tag{3}$$

$$S_{C_{2+}^=} = \frac{\sum S_i^=}{\sum (N_i \times n_i)} \times 100\% \tag{4}$$

where $CO_{inlet}$ and $CO_{outlet}$ are the moles of CO at the inlet and outlet, respectively. $S_i$ refers to the selectivity of product i on a carbon basis, $N_i$ is the molar fraction of product i, $n_i$ is the carbon number of product i and $S_i^=$ is the selectivity of olefin while the carbon number is i. WHSV is the weight hourly space velocity (mL·$g_{cat}^{-1}$·$h^{-1}$). $C_{co}$ refers to the molar concentration of CO in the feedstock, and $\xi_{Fe}$ refers to the true Fe loading measured by ICP–AES.

## Ab initio nanoreactor simulations

Ab initio nanoreactor simulations were conducted with a custom-developed version of the Vienna Ab initio Simulation Package (VASP).[34,35] Similar to the original ab initio nanoreactor proposed by Martinez et al., the motion of the molecules was restricted to a cuboidal region above the slab model by a time-dependent boundary potential $V(r, t)$, which served as a virtual piston, where $r$ is the distance of the boundary

plane above the surface and $t$ is the time. With this modified boundary potential, which oscillates between $r_1$ and $r_2$, the gas molecules undergo periodic compression and expansion, thereby enhancing the probability of collision and boosting the efficiency of reaction sampling.

$$V(r,t) = f(t)U(r,r_1,k_1) + (1-f(t))U(r,r_2,k_2) \tag{5}$$

$$U(r,r_0,k) = \frac{mk}{2}(r-r_0)^2 \theta(r-r_0) \tag{6}$$

$$f(t) = \theta\left(\left\lfloor \frac{t}{T} \right\rfloor - \frac{t}{T} + \frac{\tau}{T}\right) \tag{7}$$

Here, $k_1 = 1.0$ a.u., $r_1 = 16.0$ Å, $k_2 = 0.25$ a.u., $r_2 = 10$ Å, $\tau = 0.02$ ps, and $T = 0.25$ ps. $\lfloor \rfloor$ is the floor function, and $\theta$ is the Heaviside step function. $f(t)$ is a rectangular wave that oscillates between one (duration $\tau$) and zero (duration $T-\tau$), and $U(r, r_0, k)$ is a radial potential that is zero inside the prescribed radius $r_0$ and harmonic outside. The rectangular waveform switches the restraining potential between $U(r, r_1, k_1)$ and $U(r, r_2, k_2)$, which forces the atoms with a height position $10.0 < r < 16.0$ Å toward the surface and causes them to collide. During each compression-relaxation cycle, the molecules are propelled toward the slab surface with considerable acceleration, prompting collisions and reactions driven by an external potential. As such, the occurrence of reactions is accelerated in the ab initio nanoreactor simulations. With the degree of acceleration properly set, the kinetic energy of the molecules enables the low-barrier reactions to proceed, whereas the high-barrier reactions remain inhibited.

To simulate the continuous reaction of the reactants, the reactant molecules were added to the system in batches. The ab initio nanoreactor simulations were run with a time step of 0.5 fs. For each time-step, the energy and atomic forces were evaluated at DFT level using the same electronic settings as static DFT calculations. The temperature was controlled with a Langevin thermostat, with a target temperature of 500 K[36]. All the simulations were repeated 5 times with different initial configurations to ensure the consistency of the results.

For both ab initio nanoreactor simulations and static DFT calculations, the ε-$Fe_2C$ catalyst was modeled with ε-$Fe_2C$ (001) slab consisting of three iron layers and three carbon layers with a p (3 × 3) surface unit cell (Supplementary Fig. 21). The bottom one iron layer and two carbon layers were fixed in their bulk positions, while the top two iron layers and one carbon layer were allowed to relax. The vacuum space was set to be 20 Å. In order to study the effect of MnO on Fischer-Tropsch synthesis, an ε-$Fe_2C$ (001) slab with loaded MnO clusters was constructed. To match the experimental observations, the atomic configuration of the $Mn_4O_4$ cluster maintains the NaCl-type structure of MnO, as indicated by XPS analysis. This cluster was extracted directly from the bulk MnO structure and underwent local geometry optimization. To account for potential electronic interactions and synergistic effects among adjacent MnO clusters, two MnO clusters were placed on the surface, each containing four Mn atoms and four O atoms. The starting configurations for ab initio nanoreactor simulations were generated using the Packmol program[37]. The molecules were packed randomly in a cuboid volume above the slab model with a distance tolerance of 2.0 Å.

## Static DFT calculations

All the DFT calculations were performed with VASP[34,35]. The Perdew–Burke–Ernzerhof (PBE)[38] form of the generalized gradient approximation (GGA)[39] was used to treat the exchange-correlation energy. The electron-ion interactions were described by the projector augmented wave (PAW) method[40,41] with a plane-wave cutoff energy of 400 eV. Electron smearing was employed via a second-order

Methfessel–Paxton technique[42]. The DFT + U method was used with a U-J value of 6.0 eV. The Brillouin zone was sampled with a Monkhorst-Pack k-point grid of $2 \times 2 \times 1$[43]. All calculations were conducted with an electronic self-consistent iteration of $10^{-4}$ eV and a force tolerance of 0.02 eV Å$^{-1}$. All transition states were calculated with the climbing image nudged elastic band (CI-NEB) method[44], and vibrational frequencies were analyzed to ensure a transition state with a single imaginary frequency along the reaction coordinate.

## Data availability
The data that support the findings of this study are available within the paper and its Supplementary Information, and all data are also available from the corresponding authors upon request. Source code of NMD method is available from the corresponding authors upon request

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

## Acknowledgements

The authors are grateful for the financial support from the National Science Fund for Distinguished Young Scholars of China (Grant Nos. 22225206 to X.W. and 22025804 to Y.Y.), National Natural Science Foundation of China (22172183 to X.W.L and 22372187 to X.C.L.), CAS Project for Young Scientists in Basic Research (YSBR-005 to X.W. and X.C.L.), Key Research Program of Frontier Sciences CAS (ZDBS-LY-7007 to X.W. and X.C.L.), Major Research Plan of the National Natural Science Foundation of China (92045303 to X.W.), National Key R&D Program of China (No. 2022YFA1604103 to X.W., 2022YFB4101201 to Y.Y. and 2022YFA1604102 to X.C.L.), Informatization Plan of the Chinese Academy of Sciences (Grant No. CAS-WX2021SF0110 to X.W. and X.C.L.), Key R&D Program of Ordos (No. YF20232316 to X.W.L) and Youth Innovation Promotion Association CAS (2020179 to X.C.L.) and the funding support from Synfuels China, Co. Ltd.

## Author contributions

F.Q., J.B., and Y.C. equally contributed to this work. Y.L., Y.Y., X.W., and X.W.L. designed the study. F.Q. performed most of the reactions and sample characterization. X.W.L. carried out the X-ray structure characterization and analysis. Y.C. performed the electron microscopy characterization. J.B., H.Y., and X.C.L. performed the DFT calculations. F.Q., X.C., X.C.L., X.W.L., X.W., and D.M. wrote and revised the paper. All the authors discussed the results and commented on the manuscript. Correspondence and requests for materials should be addressed to X.C.L., X.W.L., or X.W.

## Competing interests

The authors declare no competing interests.
