## [Peer Review File · Nature Communications]

Stabilized ϵ -Fe₂C Catalyst with Mn Tuning to Suppress C₁ Byproduct Selectivity for High-Temperature Olefin SynthesisReviewers' comments:

Reviewer #1 (Remarks to the Author):

This manuscript reported the successful synthesis of highly stable Mn-modified ϵ -Fe₂C catalysts through a N-induced strategy utilizing pyrolysis of Prussian blue analogs. This catalyst showed high olefin selectivity and low C₁ product selectivity of 11.9% CO₂ and 7.1% CH₄. This catalyst demonstrated high stability at 2 MPa and 300 °C for more than 120 h. However, the manuscript is deficient in the following major aspects, which severely restricts its publication in the present form on Nature Communications.

1. The readers are more interested in the role of Mn in stabilizing the ϵ -Fe₂C phase. However, the authors did not deal with this key point.
2. As to theoretical calculations, the catalyst model should be solidly established on the basis of the characterization results, which is not the case in the present work. For example, the model for MnO cluster is arbitrary.
3. The bulk compositions and microstructures of the catalysts are lacking. For example, what are the positions of Mn and N? Since there are some N atoms in the catalysts, they may play an important role in the catalytic performance and stability.

Reviewer #2 (Remarks to the Author):

This work is a commendable effort towards advancing the field of syngas conversion to olefins, focusing on the development of an ϵ -Fe₂C catalyst modified with Mn to enhance olefin selectivity while suppressing undesired CO₂ formation. The authors of the manuscript employed a novel nitrogen-induced strategy using Prussian blue analogs as precursors, which is both innovative and promising for the sustainable production of olefins. The significant reduction in CO₂ byproduct formation to 11.9%, if reliably achieved, marks a considerable advancement for cost-effective and industrially viable iron-based FTS catalysts. The finding that Mn addition not only enhances olefin selectivity but also plays a crucial role in suppressing the WGS reaction, thereby reducing CO₂ byproduct formation, is particularly noteworthy. This study thus represents a significant step forward in addressing the longstanding challenge of achieving high olefin selectivity alongside reduced C₁ byproduct formation in syngas conversion processes.

However, there are several aspects of the manuscript that I believe require further clarification and revision to fully support the conclusions drawn before it is published. Overall, I believe that after addressing these points, the manuscript would be significantly strengthened. The work undoubtedly contributes valuable insights to the field of catalysis and syngas conversion, and with the suggested revisions, I would recommend this manuscript for publication.

Major Points:

1. The transition from Fe₄[Fe(CN)₆]₃ to Fe₂[Fe(CN)₆], and its subsequent conversion to ϵ -Fe₂N and ϵ -Fe₂C, is monitored by in situ XRD. Could the authors elaborate on the resolution and limitations of in situ XRD in distinguishing between closely related phases, especially considering the challenge in differentiating ϵ -Fe₂N from ϵ -Fe₂C? Regarding the comparative study on the thermal stability of ϵ -Fe₂C synthesized from different methods, what specific criteria were used to determine the onset of carbon depletion and phase transitions? How was the temperature ramp rate decided, and could it influence the observed stability?
2. The manuscript mentions that Mössbauer spectroscopy allows for straightforward differentiation between ϵ -Fe₂N and ϵ -Fe₂C. Could the authors provide more details on the parameters and calibration methods used in MES to ensure the reliability of these measurements?
3. The proposed contribution of nitrogen atoms to the thermal stability of ϵ -Fe₂C is intriguing. Have preliminary computational studies or literature precedents been considered to support this hypothesis, and how might they guide future research?
4. The enhancement of olefin selectivity through Mn promotion is a critical finding of this study.

However, the manuscript could provide a deeper analysis of the ethene and propene pulse experiments to elucidate how Mn affects the hydrogenation process of olefins more explicitly. Clarifying this mechanism would enhance the overall understanding of the catalytic process and the role of Mn. While the positive impact of Mn on olefin selectivity and C1 byproduct suppression is clear, the manuscript could benefit from a more nuanced discussion on the limitations or potential trade-offs associated with Mn addition. For instance, does Mn incorporation affect the catalyst's long-term stability or its activity for other important reactions in FTS?

5. The manuscript briefly compares the ϵ -Fe₂C-xMn catalysts' performance with previously reported catalysts. Could the authors expand on this comparison, perhaps including a discussion on the cost-effectiveness, scalability, and environmental impact of their synthesis method versus others?

6. While the manuscript mentions catalytic stability, there is less focus on potential deactivation mechanism. Could the authors comment on any observed or anticipated catalyst deactivation phenomena over longer reaction periods?

Minor comments and suggestions:

1. In Figure 2b, the superscript "TW" in the catalyst names should be noted in the legend.

2. In Figure 4f, the pathway for CO₂ generation could also proceed through the formate or carbonate mechanism via the reaction of CO with surface OH species. The authors should discuss the possibility of these pathways.

3. The authors should ensure uniformity in the format of all figures and tables in the supplementary files, paying special attention to font style and whether or not the titles are bolded.

4. In Supplementary File Figure 7, the scale bar lengths for 200 nm in images (a) and (b) appear inconsistent. The authors are requested to verify if there is an error.

5. The authors should provide the calculation formula for the C₂+ olefin selectivity and present the carbon balance of the reaction.

Reviewer #3 (Remarks to the Author):

In this work, the authors presented ϵ -Fe₂C catalysts for the conversion of synthesis gas to olefins. They highlighted their N-induced catalyst preparation strategy which led to superior ϵ -Fe₂C stability. In addition to catalyst stability, the addition of Mn promoter improved selectivity towards C₂+ hydrocarbon products. The origin of the Mn promotion effect is investigated experimentally and theoretically. The quality of the experimental and theoretical work is high, however in my view, the novelty/ advancement is not sufficient to warrant its publication in Nature Communication. I present my case as follows:

1. The superior performance and strategies to stabilise ϵ -Fe₂C catalysts are well documented (ref. 21 – 24). The N-induced strategy to stabilise ϵ -Fe₂C catalysts has been reported previously (ref. 24).

What could perhaps be more interesting and significant is a deeper understanding of the N-induced stability enhancement, however the authors consider this task to be outside the scope of this work.

2. The authors carburized α -Fe directly to use as a reference ϵ -Fe₂C catalyst and showed that their N-induced ϵ -Fe₂C catalyst was more stable. However, this is not a good reference to demonstrate significant improvement to state-of-art. If the authors could demonstrate and explain why their N-induced catalyst preparation method result in the most selective and stable ϵ -Fe₂C catalysts in comparison to other state-of-art ϵ -Fe₂C catalysts (ref 21 to 24), they would have a stronger case for a publication in Nature Communication.

3. The authors claimed that their catalysts showed suppressed C₁ products in comparison to state-of-art literature, as illustrated in Figure 2C. However, the results of ref. 21 to 24 are missing from the comparison in Figure 2C, and those results showed comparable or better suppression of C₁ products. Hence, the reviewer is not convinced by this claim for significant novelty.

4. This study delved deeply into the impact of Mn, a widely recognized promoter. The results obtained align closely with existing literature findings.

Response to Reviewers:

We express our sincere gratitude to the reviewers for their valuable and constructive comments on our manuscript. We have addressed each comment in detail and have revised the manuscript thoroughly, considering all the feedback and suggestions. In this response letter, the reviewers' comments are presented in *black italics*, our responses are in blue, and all changes are marked in **RED** color in the revised manuscript and supporting information.

Reviewers' comments:

Reviewer #1

This manuscript reported the successful synthesis of highly stable Mn-modified ϵ -Fe₂C catalysts through a N-induced strategy utilizing pyrolysis of Prussian blue analogs. This catalyst showed high olefin selectivity and low C1 product selectivity of 11.9% CO₂ and 7.1% CH₄. This catalyst demonstrated high stability at 2 MPa and 300 °C for more than 120 h. However, the manuscript is deficient in the following major aspects, which severely restricts its publication in the present form on Nature Communications.

1. The readers are more interested in the role of Mn in stabilizing the ϵ -Fe₂C phase. However, the authors did not deal with this key point.

Author reply: We appreciate the reviewer's speculation on whether Mn stabilizes the ϵ -Fe₂C phase. In our original manuscript, we provided data from XRD and Mössbauer spectroscopy analyses of the ϵ -Fe₂C-xMn catalyst series after FTS (**Fig. 1e, 1f and Supplementary Table 5**), showing that all catalysts maintained the Fe₂C phase without forming any other iron oxides or carbides, irrespective of the presence of Mn. This sufficiently demonstrates that Mn do not significantly affect the stability of the ϵ -Fe₂C phase.

To further verify the impact of Mn on the stability of ϵ -Fe₂C, we have added *in-situ* XRD of the ϵ -Fe₂C-0.4Mn catalyst after FTS under a temperature-programmed heating in a helium atmosphere (**Fig. R1**). We observed that the ϵ -Fe₂C phase begins to transform at a temperature close to 440 °C, similar to the transformation temperature of ϵ -Fe₂C without Mn promoter (**Fig. 1a**), further confirming that Mn does not affect the thermal stability of the ϵ -Fe₂C phase. For clarity in our revised manuscript, we have explicitly stated that Mn does not influence the thermal stability of the ϵ -Fe₂C phase (see Page 9 line221-224) and have added **Fig. R1** as **Supplementary Fig. 10** in the

supplementary files.

Fig. R1 *In situ* XRD patterns of temperature-programmed phase transition of ϵ - Fe_2C -0.4Mn sample in a helium atmosphere from room temperature to 650 °C. Conditions: He 50 $\text{mL}\cdot\text{min}^{-1}$, 5 $^\circ\text{C}\cdot\text{min}^{-1}$

2. As to theoretical calculations, the catalyst model should be solidly established on the basis of the characterization results, which is not the case in the present work. For example, the model for MnO cluster is arbitrary.

Author reply: We sincerely appreciate the reviewer's thorough review and valuable comments regarding the selection of the MnO cluster for theoretical calculations. We acknowledge the critical importance of aligning our theoretical model with experimental characterizations to ensure the relevance and accuracy of our computational findings. Our choice to utilize the aforementioned supported MnO cluster as our model was guided by several key considerations.

Our experimental results, including XRD as shown in **Fig.1e**, XPS in **Fig. 3d**, and XAS in **Fig. 3b and 3c**, confirm the presence of Mn in the form of MnO. Comparative analysis of element distribution by inductively coupled plasma (ICP) for bulk analysis and XPS for surface characterization (**Fig. 3a**) suggests that MnO preferentially localizes on the surface of the catalysts. This is further supported by EDS mapping, which confirms the high dispersion of MnO clusters on the Fe_2C surface (**Fig. 3e**). Additionally, the coordination number provided in **Supplementary Table 13** indicates that these MnO entities are not isolated atoms but rather clusters $(\text{MnO})_n$, emphasizing their clustered structure. Nevertheless, the precise quantification of atoms and detailed atomic configuration of the MnO clusters remain undetermined due to the limitations

of the current characterization techniques.

Second, due to the computational constraints of density functional theory (DFT), particularly *ab initio* molecular dynamics (AIMD) simulations, we modeled the Fe₂C-supported MnO particle using a smaller MnO cluster (Mn₄O₄), despite the actual number of atoms being unknown experimentally. This modeling approach aligns with standard practices in the investigation of similar supported particles as documented in scientific literature (*Nat. Catal.* **2022**, 5, 1051; *Nat. Commun.* **2021**, 12, 5770; *Nat. Commun.* **2019**, 10, 954). To match the experimental observations, the atomic configuration of the Mn₄O₄ cluster maintains the NaCl-type structure of MnO, as indicated by XPS analysis. This cluster was extracted directly from the bulk MnO structure and underwent local geometry optimization. The model was stable enough and did not deform during the optimization. We refrained from employing further global optimization or Grand Canonical Monte Carlo simulations, as these methods could yield configurations inconsistent with the NaCl-type structure confirmed by XPS.

Thirdly, experimental analysis suggests a high density of MnO clusters on the Fe₂C surface, implying proximity between these clusters. To account for potential electronic interactions and synergistic effects among adjacent MnO clusters, we incorporated two Mn₄O₄ clusters on the Fe₂C surface in our model. Our AIMD simulations confirm that these neighboring MnO clusters indeed exhibit a synergistic effect, particularly in the adsorption of CO molecules. This interaction significantly stabilizes the adsorption of CO between the clusters.

Overall, our theoretical model selection is robustly justified by the highest-quality experimental characterizations available. The supported Mn₄O₄ clusters not only mirrors the experimentally observed form of manganese but also fits well within the constraints of our computational resources. We have included a detailed justification of the theoretical model for the MnO/ ϵ -Fe₂C catalyst in the revised manuscript. (see Page 23, line 582-597).

3. The bulk compositions and microstructures of the catalysts are lacking. For example, what are the positions of Mn and N? Since there are some N atoms in the catalysts, they may play an important role in the catalytic performance and stability.

Author reply: Thank the reviewer for your comments. We are surprised by the reviewer's concerns regarding an alleged lack of characterization of the bulk compositions and microstructures of our catalysts. In the original version, we have

employed a comprehensive suite of characterization techniques including *in situ* XRD, MES, EXAFS, XANES, ICP, XPS, HRTEM, and EDS mapping. These methods were systematically used to analyze the bulk compositions (XRD, MES, XANES), valence states of Fe and Mn (XPS, MES, EXAFS), element analysis (ICP, XPS, Mapping), and microstructures (HRTEM).

Most importantly, the distribution of Mn on the catalyst, a key focus of our work, has been thoroughly investigated. Our results confirm the presence of Mn as MnO, as evidenced by XRD (**Fig. 1e**), XPS (**Fig. 3d**), and XAS (**Figs. 3b and c**). Comparative analyses by ICP for bulk and XPS for surface characterization (**Fig. 3a**) indicate that MnO is preferentially localized on the surface of the catalysts. This is further supported by EDS mapping, which shows a high dispersion of MnO clusters on the ϵ -Fe₂C surface (**Fig. 3e**). Additionally, we have added the coordination numbers in **Supplementary Table 13**, which demonstrate that these MnO entities are clustered, not isolated atoms, highlighting their structured aggregation.

Indeed, characterizing trace light elements such as hydrogen (H) and nitrogen (N) in catalysts has always posed a challenge. In response to the reviewer's queries, we have endeavored to characterize the position of N atoms. Through STEM-EDS and XPS characterization results (**Fig. R2**), we concluded that N atoms predominantly reside inside the catalysts, with no presence of N on the surface. Concurrently, STEM-EDS quantitative analysis confirmed the minimal content of N, with an N/Fe ratio of merely about 2.5% (**Table R1**). As early as the 1950s, Anderson R. B. and colleagues conducted systematic studies using iron nitride as FTS catalysts (*JACS*, **1950**, 72, 3502; *JACS*, **1953**, 75, 1442.). Building upon this foundation, our work has further developed and systematically compared the FTS reactivity of ϵ -Fe₂N, ϵ -Fe₃N, and Fe₄N at lower temperature about 240 °C (*ChemCatChem* **2020**, 12, 1939). Our findings reveal that after an initial loss of nitrogen, a significant amount of carbonitrides forms. These carbonitrides slowly lose nitrogen as the reaction progresses, with carbon atoms replacing nitrogen at the octahedral sites in the hcp Fe matrix. Given that our reaction temperature reached up to 300°C, only a small fraction of nitrogen, corresponding to an N/Fe ratio of about 2.5%, remains.

The presence of trace N atoms plays a vital role in the stability of Fe₂C. By comparing *in situ* XRD data (**Fig. 1a and 1b**), it is evident that the stability of Fe₂C containing trace N atoms is substantially enhanced, with a notable delay in the phase transition temperature from 330 to 440 °C. To further elucidate the effect of N on catalyst stability, we have performed DFT calculations to assess its influence on the

catalyst's formation energy. These calculations indicate that the incorporation of N lowers the formation energy (**Fig. R3**), thereby augmenting the catalyst's stability. Even under conditions of elevated temperature, pressure, and CO partial pressure, where the N concentration tends to diminish, it still markedly improves the stability of our catalyst.

Considering the impact of N atoms on catalytic performance, we have conducted comparative evaluation experiments. The product distribution across different activation condition catalysts showed no marked difference (Supplementary Table 3). However, these atmospheres led to significantly reduced FTS reaction stability, accompanied by phase transformation of the catalysts. (Supplementary Fig. 3c and Supplementary Table 3). We have incorporated **Fig. R2**, **Table R1** and **Fig. R3** into the Supplementary Information as **Supplementary Fig. 4**, **Supplementary Table 4**, and **Supplementary Fig. 5** respectively. The above statements have been added to the revised manuscript (page 6, line 156 -167).

Fig. R2 X-ray energy dispersive spectroscopy image (a-c) and (d) XPS profiles in the N 1s of $\epsilon\text{-Fe}_2\text{C}$.

Table R1 The ratio of N/Fe from quantitative analysis results in STEM-EDS

Sample	$\epsilon\text{-Fe}_2\text{C-spent}$	$\epsilon\text{-Fe}_2\text{C-0.06Mn-spent}$	$\epsilon\text{-Fe}_2\text{C-0.4Mn-spent}$
Ratio of N/Fe (%)	2.2	2.5	2.6

Fig. R3. (a) Formation energies predicted by Cluster Expansion method for 2868 different N configurations. (b) The relationship between predicted energies and DFT calculation energies.

Reviewer #2

This work is a commendable effort towards advancing the field of syngas conversion to olefins, focusing on the development of an ϵ -Fe₂C catalyst modified with Mn to enhance olefin selectivity while suppressing undesired CO₂ formation. The authors of the manuscript employed a novel nitrogen-induced strategy using Prussian blue analogs as precursors, which is both innovative and promising for the sustainable production of olefins. The significant reduction in CO₂ byproduct formation to 11.9%, if reliably achieved, marks a considerable advancement for cost-effective and industrially viable iron-based FTS catalysts. The finding that Mn addition not only enhances olefin selectivity but also plays a crucial role in suppressing the WGS reaction, thereby reducing CO₂ byproduct formation, is particularly noteworthy. This study thus represents a significant step forward in addressing the longstanding challenge of achieving high olefin selectivity alongside reduced C1 byproduct formation in syngas conversion processes.

However, there are several aspects of the manuscript that I believe require further clarification and revision to fully support the conclusions drawn before it is published. Overall, I believe that after addressing these points, the manuscript would be significantly strengthened. The work undoubtedly contributes valuable insights to the field of catalysis and syngas conversion, and with the suggested revisions, I would recommend this manuscript for publication.

Author reply: We sincerely appreciate your comprehensive review and the insightful comments on our manuscript. We are particularly grateful for your recognition of the innovative aspects and the potential impact of our study, which employs Mn-modified ϵ -Fe₂C catalysts for syngas conversion.

Our work contributes valuable insights to the field of catalysis and syngas conversion, and we believe that with the suggested revisions, the manuscript will be well-positioned for publication. We look forward to making the necessary enhancements to our study and appreciate your recommendation for publication following these revisions.

Major Points:

1. The transition from Fe₄[Fe(CN)₆]₃ to Fe₂[Fe(CN)₆], and its subsequent conversion to ϵ -Fe₂N and ϵ -Fe₂C, is monitored by in situ XRD. Could the authors elaborate on the resolution and limitations of in situ XRD in distinguishing between closely related

phases, especially considering the challenge in differentiating ϵ -Fe₂N from ϵ -Fe₂C? Regarding the comparative study on the thermal stability of ϵ -Fe₂C synthesized from different methods, what specific criteria were used to determine the onset of carbon depletion and phase transitions? How was the temperature ramp rate decided, and could it influence the observed stability?

Author reply: Thank you for your valuable comment. Your concerns about the resolution and limitations of *in situ* XRD in distinguishing closely related phases, as well as the specific criteria used for the comparative study on thermal stability, are well-taken. Firstly, regarding the transition from Fe₄[Fe(CN)₆]₃ to Fe₂[Fe(CN)₆], and its subsequent conversion to ϵ -Fe₂N and ϵ -Fe₂C (**Fig. 1a** and **Supplementary Fig. 1**), we monitored these changes using *in situ* XRD. While *in situ* XRD is a powerful technique for studying phase transitions, it does have limitations in resolving closely related phases, especially when dealing with similar crystal structures. In our case, differentiating ϵ -Fe₂N from ϵ -Fe₂C can indeed be challenging due to their structural similarities. To address this, we carefully analyzed the diffraction patterns, taking into account peak positions, intensities, and changes in peak shapes. We also complemented our XRD data with other characterization techniques, such as Mossbauer spectroscopy, to provide additional evidence for phase identification.

For the comparative study on the thermal stability of ϵ -Fe₂C synthesized from different methods, we used specific criteria to determine the onset of carbon depletion and phase transitions. These criteria included changes in the XRD patterns, such as the appearance or disappearance of certain peaks, as well as shifts in peak positions that indicated lattice parameter changes. The temperature ramp rate was carefully chosen based on our experimental setup and the kinetics of the phase transitions we were studying. While the ramp rate can influence the observed stability, we ensured that it was consistent across all samples and methods to allow for a fair comparison.

2. The manuscript mentions that Mossbauer spectroscopy allows for straightforward differentiation between ϵ -Fe₂N and ϵ -Fe₂C. Could the authors provide more details on the parameters and calibration methods used in MES to ensure the reliability of these measurements?

Author reply: Thank the reviewer for your suggestion. We have provided detailed information on the Mossbauer spectroscopy fitting methods and parameters in the

Experimental Methods section and in Supplementary Tables 1, 2, 5, and 6. Additionally, we have referred to literature on Mossbauer spectroscopic studies of Fe₂N and Fe₂C (*Hyperfine. Interact.* **1994**, 94, 2067.). Furthermore, our previous work on various iron carbides (*Sci. Rep.* **2016**, 6, 26184; *J. Phys. Chem. C* **2017**, 121 (39), 21390), including Fe₂C, has laid a solid foundation for accurately identifying ϵ -Fe₂N and ϵ -Fe₂C.

3. The proposed contribution of nitrogen atoms to the thermal stability of ϵ -Fe₂C is intriguing. Have preliminary computational studies or literature precedents been considered to support this hypothesis, and how might they guide future research

Author reply: Through EDS and XPS characterization results (**Fig. R2**), we concluded that N atoms predominantly reside inside the catalysts, with no presence of N on the surface. Concurrently, STEM-EDS quantitative analysis confirmed the minimal content of N, with an N/Fe ratio of merely about 2.5% (**Table R1**). As early as the 1950s, Anderson R. B. and colleagues conducted systematic studies using iron nitride as FTS catalysts. Building upon this foundation, our work has further developed and systematically compared the FTS reactivity of ϵ -Fe₂N, ϵ -Fe₃N, and Fe₄N at lower temperature about 240 °C (*ChemCatChem* **2020**, 12, 1939). Our findings reveal that after an initial loss of nitrogen, a significant amount of carbonitrides forms. These carbonitrides slowly lose nitrogen as the reaction progresses, with carbon atoms replacing nitrogen at the octahedral sites in the hcp Fe matrix. Given that our reaction temperature reached up to 300°C, only a small fraction of nitrogen, corresponding to an N/Fe ratio of about 2.5%, remains.

The presence of trace N atoms plays a vital role in the stability of Fe₂C. By comparing *in situ* XRD data (**Fig. 1a and 1b**), it is evident that the stability of Fe₂C containing trace N atoms is substantially enhanced, with a notable delay in the phase transition temperature from 330 to 440 °C. To further elucidate the effect of N on catalyst stability, we have performed DFT calculations to assess its influence on the catalyst's formation energy. These calculations indicate that the incorporation of N lowers the formation energy (**Fig. R3**), thereby augmenting the catalyst's stability. Even under conditions of elevated temperature, pressure, and CO partial pressure, where the N concentration tends to diminish, it still markedly improves the stability of our catalyst. The above statements have been added to the revised manuscript (page 6, line 156-167).

4. The enhancement of olefin selectivity through Mn promotion is a critical finding of this study. However, the manuscript could provide a deeper analysis of the ethene and propene pulse experiments to elucidate how Mn affects the hydrogenation process of olefins more explicitly. Clarifying this mechanism would enhance the overall understanding of the catalytic process and the role of Mn. While the positive impact of Mn on olefin selectivity and C1 byproduct suppression is clear, the manuscript could benefit from a more nuanced discussion on the limitations or potential trade-offs associated with Mn addition. For instance, does Mn incorporation affect the catalyst's long-term stability or its activity for other important reactions in FTS?

Author reply: We appreciate the insightful comment from the reviewer, which highlighted the unique novelty of our work. The Mn promoter is crucial in achieving the remarkable olefin selectivity and C1 byproduct suppression observed in the ϵ -Fe₂C/Mn catalyst. Understanding the synergy between ϵ -Fe₂C and Mn promotion is key to guiding the design of more effective and sustainable catalysts for olefin production, and therefore, it is the main focus of this work.

Regarding CO₂ suppression, our mechanistic study shows that Mn promotion effectively suppresses both the primary and secondary pathways of CO₂ formation. The suppression of the primary pathway is evident from the CO₂ selectivity in both native and Mn-promoted ϵ -Fe₂C catalysts as a function of CO conversion (**Fig. 4a**), as reported in previous studies (*J. Catal.* **2010**, 272, 287; *Catal. Sci. Technol.* **2018**, 8, 5288). The suppression of the secondary pathway is demonstrated by the reduced CO₂/CO intensity observed in the WGS pulse reactions (**Fig. 4b**) and the temperature-programmed surface reaction (**Supplementary Fig. 15**). Furthermore, *ab initio* nanoreactor simulations show that CO molecules tend to migrate towards and become trapped near the supported MnO clusters, thereby hindering their reaction with surface O* to form CO₂ and suppresses both primary and secondary CO₂ formation. Additionally, static calculations indicate that the CO₂ formation barrier increases by more than 1 eV with Mn promotion.

Regarding the extremely high selectivity of olefins relative to paraffins, detailed analysis from ethene and propene pulse experiments indicates that Mn promotion significantly suppresses the secondary hydrogenation of olefins. DFT studies reveal that the adsorption energy of ethene decreases dramatically from -1.50 eV to -0.67 eV with Mn promotion (**Supplementary Fig. 16**), which greatly facilitates their desorption from the surface and helps avoid their secondary hydrogenation to paraffins through re-

adsorption. Furthermore, additional barrier calculations indicate that the formation of C_2H_6 is kinetically suppressed on $MnO/\epsilon-Fe_2C(001)$ compared to $\epsilon-Fe_2C(001)$ without Mn (2.38 eV vs. 1.64 eV see **Fig. R4**), significantly enhancing the selectivity towards ethylene. This improvement is largely attributable to the reduced activation of ethene molecules due to weaker adsorption on the Mn-promoted surface. Overall, the mechanistic study outlined above provides a solid rationale for the remarkable olefin selectivity and C1 byproduct suppression observed in the $\epsilon-Fe_2C$ -Mn catalyst.

Fig. R4 The free energy surface for propene hydrogenation into propane.

Regarding the limitations or potential trade-offs associated with Mn addition, we observed a slight decrease in CO conversion as Mn loading increased. This reduction may be attributed to the reduced exposure to the active site, which can become covered by the highly dispersed MnO clusters. However, the incorporation of Mn did not affect the long-term stability of the catalyst. As shown in the **Fig. R1** and **Fig. 1a**, the introduction of the Mn promoter did not alter the stability of the $\epsilon-Fe_2C$ phase. Furthermore, stability tests under reaction condition confirmed that $\epsilon-Fe_2C$ remains stable (**Fig. 2a and 2c, Supplementary Fig. 3c**).

5. The manuscript briefly compares the $\epsilon-Fe_2C$ -xMn catalysts' performance with previously reported catalysts. Could the authors expand on this comparison, perhaps including a discussion on the cost-effectiveness, scalability, and environmental impact of their synthesis method versus others?

Author reply: Thank you for your valuable comments on our manuscript. We

appreciate your suggestion to expand the comparison of the ϵ -Fe₂C-xMn catalysts' performance with previously reported catalysts. The raw material cost price of ϵ -Fe₂C-xMn catalysts is approximately 4 U.S. dollar per gram. While other reported catalysts vary from 0.5 - 10 U.S. dollar per gram (Nature. 2016, 538, 84-87; Angew. Chem. Int. Ed. 2016, 55, 9902-9907; Science. 2012, 335, 835-838; Science. 2021, 371, 610-613; Nat. Nanotechnol. 2022, 17, 714-720; Science. 2022, 377, 406-410; Science. 2016, 351, 1065-1068). Our catalyst is prepared by coprecipitation, which is easily scale up preparation and requires no additional energy input. Furthermore, we assess the environmental impact of our synthesis method, the solvent is water and no other harmful substances are produced including the generation of waste and potential emissions.

6. While the manuscript mentions catalytic stability, there is less focus on potential deactivation mechanism. Could the authors comment on any observed or anticipated catalyst deactivation phenomena over longer reaction periods?

Author reply: Thank the reviewer for your inquiry regarding the potential deactivation mechanisms of Fe₂C FTS catalysts. The potential causes for catalyst deactivation are multifaceted and can include:

1. **Oxidation of the Active Phase:** The ϵ -Fe₂C-Mn catalyst may undergo oxidation by H₂O during the FTS reaction, since the lower CO₂ selectivity, leading to a decrease in catalytic performance.
2. **Coking:** Given that ϵ -Fe₂C is a carbon-rich phase among iron carbides, there is a tendency for lattice carbon atoms to migrate to the surface and form graphite or amorphous carbon structures. Furthermore, the aggregation of CH_x species on the catalyst surface, coupled with the progressive removal of hydrogen, may result in carbonaceous deposits. These deposits can obscure the catalyst surface, impeding interaction with reactants and ultimately causing deactivation.
3. **Sintering of the Active Phase:** At high temperatures, the ϵ -Fe₂C catalyst may sinter, resulting in a reduced effective surface area and lower catalytic activity.

These are the primary reasons we proposed for the potential deactivation of ϵ -Fe₂C FTS catalysts.

Minor comments and suggestions:

1. In Figure 2b, the superscript "Tw" in the catalyst names should be noted in the legend.

Author reply: Thank you for your inquiry about the notation used in Figure 2b. The superscript "TW" stands for "this work," We have explicitly added the clarification that 'TW' stands for 'This Work' in the legend of the revised manuscript."

2. In Figure 4f, the pathway for CO₂ generation could also proceed through the formate or carbonate mechanism via the reaction of CO with surface OH species. The authors should discuss the possibility of these pathways.

Author reply: Thank you for your valuable comment. We calculated the barrier of reaction (CO* + OH* → COOH) in ε-Fe₂C (001) and ε-Fe₂C (001)-MnO surface, which are 1.15 and 2.41 eV, respectively (**Fig. R5** as **Fig. 4f** in revised version). The results demonstrated that the MnO cluster also inhibited the reaction of CO + OH.

Fig. R5 The potential energy surface of H₂O dissociation, OH dissociation, CO + O and CO + OH.

3. The authors should ensure uniformity in the format of all figures and tables in the supplementary files, paying special attention to font style and whether or not the titles are bolded.

Author reply: Thank you for your constructive comment regarding the formatting consistency of figures and tables in the supplementary files. We appreciate your attention to detail. We have reviewed all the supplementary materials and ensured uniformity in the font style and formatting of titles across all figures and tables. The

titles have now been uniformly bolded to enhance clarity and coherence. We believe these revisions improve the presentation and readability of our supplementary data.

4. In Supplementary File Figure 7, the scale bar lengths for 200 nm in images (a) and (b) appear inconsistent. The authors are requested to verify if there is an error.

Author reply: We have carefully re-examined the scale bars to make sure there is no error.

5. The authors should provide the calculation formula for the C₂⁺ olefin selectivity and present the carbon balance of the reaction.

Author reply: We have revised manuscript to clarify the calculation formula for the C₂₊ olefin selectivity and present the carbon balance of the reaction.

Reviewer #3:

In this work, the authors presented ϵ -Fe₂C catalysts for the conversion of synthesis gas to olefins. They highlighted their N-induced catalyst preparation strategy which led to superior ϵ -Fe₂C stability. In addition to catalyst stability, the addition of Mn promoter improved selectivity towards C₂⁺ hydrocarbon products. The origin of the Mn promotion effect is investigated experimentally and theoretically. The quality of the experimental and theoretical work is high, however in my view, the novelty/ advancement is not sufficient to warrant its publication in Nature Communication. I present my case as follows:

Author reply: We are deeply grateful for the reviewer's high regard for the quality of our work. We understand your concerns regarding the novelty and advancement of our work. Your skepticism provides us with an opportunity to further clarify the unique value and contributions of our research.

The existing literature on ϵ -Fe₂C (*ref. 21-24*) does not overshadow the innovative aspects of our work. Our research introduces significant advancements in ϵ -Fe₂C catalysts, distinguished by two main innovations:

Methodological Breakthrough: We have pioneered a novel preparation strategy for ϵ -Fe₂C via the pyrolysis of PBAs under NH₃ atmosphere and following carburization, diverging from conventional approach. Our ϵ -Fe₂C catalysts exhibit unprecedented stability at temperatures about 440 °C (**Fig. 1a**), a feat not reported in prior ϵ -Fe₂C studies (*ref. 21-24*). The remarkable stability of our ϵ -Fe₂C catalysts is probably attributed to nitrogen incorporation. Furthermore, we report for the first time the preparation of Mn-modified ϵ -Fe₂C catalysts through the pyrolysis of PBAs, which are utilized for high olefin production from syngas while simultaneously suppressing C1 byproducts.

Performance Excellence: Our ϵ -Fe₂C-Mn catalysts outperform existing state-of-the-art ϵ -Fe₂C systems by producing higher value-added olefins and suppressing C1 byproducts. A standout feature of our system is its olefin selectivity, which constitutes 70.2% of the total product yield. The ϵ -Fe₂C-Mn system not only showcases optimal olefin selectivity but also demonstrates remarkable efficiency in reducing C1 byproducts, with CO₂ selectivity at just 11.9% and overall C1 byproduct selectivity below 20%. This performance is notably superior to the 22% C1 byproduct selectivity reported by Ding *et al.* using a hydrophobic catalyst (*Science* **2021**, 371, 610.). Our results exceed both other state-of-art ϵ -Fe₂C catalysts (*ref. 21 - 24*) and recent outcomes

from syngas-to-olefin processes, as detailed in **Fig. R6**, **Table R2**, and **Supplementary Table 11**. The superior performance of our ϵ -Fe₂C-xMn catalysts, particularly in minimizing CO₂ formation, is largely attributed to the interface of dispersed MnO cluster and Fe₂C, which crucially limits CO to CO₂ conversion.

We are profoundly grateful to the reviewer for your high standards and invaluable suggestions. Following your expert guidance, we have refined the presentation of our manuscript. (see title, abstract, Page2 line 54-58, Page4 line91-101, Page5 line128-137, Page6 line156-167, Page10 line253-273, Page11 line277-286) These efforts have significantly elevated the quality of our work. We hope that these enhancements to our work collectively represent a considerable leap forward in the design and application of ϵ -Fe₂C catalysts for FTS, warranting publication in *Nature Communications*. With these improvements in mind, we kindly request a re-evaluation of our manuscript. We are confident that our findings offer substantial insights into the development of more efficient and selective catalysts for FTS processes.

Fig. R6 Comparison of the catalytic performance of ϵ -Fe₂C-xMn with that of other previously reported ϵ -Fe₂C catalysts. (a: reaction conditions: 0.10 g of catalyst, 280 °C, 2.0 MPa, H₂/CO = 2.5, GHSV = 60 L·g_{cat}⁻¹·h⁻¹, b: reaction conditions: 0.10 g of catalyst, 300 °C, 2.0 MPa, H₂/CO = 2.5, GHSV = 20 L·g_{cat}⁻¹·h⁻¹) ([1] *Nat. Commun.* 2020, 11, 6219; [2] *Appl. Catal. B: Environ.* 2021, 284, 119702; [3] *Sci. Adv.* 2018, 4, eaau2947; [4] *Nat. Commun.* 2014, 5, 5783)

Table R2 Comparison of the performance of ϵ -Fe₂C-0.4Mn in syngas-to-olefins synthesis with other ϵ -Fe₂C catalysts reported in the literature (a: The values denote the selectivity and yield of lower olefins (C₂₋₄⁼))

Catalysts	WHSV (ml·g _{cat} ⁻¹ ·h ⁻¹)	T (°C)	H ₂ /CO ratio	CO	Sel. [mol%]					Ref.
					CO ₂	CH ₄	C1	C ₂₋₄	C ₅₊ Olefins	

			Conv				Total				
ϵ -Fe ₂ C	18000	235	1.5	15.0	5.0	17.0	22.0	29.0	49.0	-	Sci. Adv. 2018, 4, eaau2947
RQ- ϵ -Fe ₂ C	-	170	2	76.0	19.0	18.0	37.0	28.4	34.6	-	Nat. Commun. 2014, 5, 5783
ϵ -Fe ₂ C@graphene	-	300	1	-	20.3	8.2	28.5	19.0	52.4	13.4 ^a	Nat. Commun. 2020, 11, 6219
ϵ -Fe ₂ C/Al ₂ O ₃	9000	280	1	-	19.2	9.0	28.2	25.3	46.5	18.0 ^a	Appl. Catal. B 2021, 284, 119702
ϵ -Fe ₂ C-0.4Mn	20000	300	2.5	41.0	11.9	7.1	19.0	42.8	38.2	70.2	This work
ϵ -Fe ₂ C-0.4Mn	60000	280	2.5	17.1	2.4	9.9	12.3	52.8	34.9	71.3	This work

1. The superior performance and strategies to stabilise ϵ -Fe₂C catalysts are well documented (ref. 21 - 24). The N-induced strategy to stabilise ϵ -Fe₂C catalysts has been reported previously (ref. 24). What could perhaps be more interesting and significant is a deeper understanding of the N-induced stability enhancement, however the authors consider this task to be outside the scope of this work.

Author reply: We sincerely appreciate the reviewer's insightful comments. The studies cited in references 21 to 24 are indeed pivotal for grasping the property of ϵ -Fe₂C phase and its exact role on FTS. We wish to elucidate and highlight the unique aspects and innovative contributions of our research concerning ϵ -Fe₂C, distinguishing it from these seminal works.

As detailed in **Table R2** and illustrated in **Fig. R6**, our ϵ -Fe₂C-xMn system demonstrates unique FTS performance, specifically in suppressing C1 by-product selectivity and enhancing olefin selectivity. Under low-temperature conditions (235 °C and 15% CO conversion), the RQ- ϵ -Fe₂C reported in Ref. 22 shows a CO₂ selectivity of 5% and a CH₄ selectivity of 17%. In contrast, our ϵ -Fe₂C-Mn operates effectively even at a higher temperature of 280 °C, achieving a markedly lower CO₂ selectivity of 2.4% and a methane selectivity of 9.9%. These results not only highlight the enhanced efficiency of our Mn-modified ϵ -Fe₂C in minimizing C1 by-products but also demonstrate its superior performance in achieving high olefin selectivity compared to the established state-of-the-art ϵ -Fe₂C catalysts reported in the literature (ref. 21 - 24). Our innovative approach focuses on the modification of ϵ -Fe₂C with Mn to tune the catalytic activity and selectivity. This modification has enabled a significant shift in catalytic performance metrics, particularly under the challenging conditions of higher temperature operation where traditional ϵ -Fe₂C catalysts typically show decreased selectivity and stability.

As early as the 1950s, Anderson R. B. and colleagues conducted systematic studies using iron nitride as FTS catalysts (*JACS*, **1950**, 72, 3502; *JACS*, **1953**, 75, 1442).

Building upon this foundation, our work has further developed and systematically compared the FTS reactivity of ϵ -Fe₂N, ϵ -Fe₃N, and Fe₄N (see *ChemCatChem* **2020**, 12, 1939). Our findings indicate that following the initial loss of nitrogen, a substantial amount of carbonitrides form, which very slowly lose their nitrogen as the reaction proceeds. Therefore, the strategy reported by Fu *et al.* (Ref.22, *Appl. Catal. B* **2021**, 284, 119702) of preparing Fe₂C/Al₂O₃ from Fe₂N/Al₂O₃ is not novel. Moreover, the support can enhance the stability of Fe₂C (*J. Phys. Chem.* **1981**, 85, 2484), and the stabilizing effect of Al₂O₃ on Fe₂C cannot be ruled out.

In response to the reviewer's advice, we have included a detailed discussion in the revised manuscript about the reasons for the N-induced stability enhancement of Fe₂C (see Page6 line156-167). Our theoretical calculations suggest that the smaller atomic radius of N compared to C results in less strain when N occupies the octahedral interstices in the hcp iron lattice. This is corroborated by our findings that replacing a small amount of carbon atoms with nitrogen in the Fe₂C lattice yields a negative formation energy, indicating that such nitrogen doping can effectively enhance the stabilization energy of Fe₂C (**Fig. R3**).

We believe that these additions and clarifications will address the concerns raised and further underscore the novelty and significance of our work. Once again, we thank you for your constructive comments, which has been instrumental in refining our study and elucidating the innovative aspects of our research.

2. The authors carburized α -Fe directly to use as a reference ϵ -Fe₂C catalyst and showed that their N-induced ϵ -Fe₂C catalyst was more stable. However, this is not a good reference to demonstrate significant improvement to state-of-art. If the authors could demonstrate and explain why their N-induced catalyst preparation method result in the most selective and stable ϵ -Fe₂C catalysts in comparison to other state-of-art ϵ -Fe₂C catalysts (ref 21 to 24), they would have a stronger case for a publication in Nature Communication.

Author reply: Thank you for the insightful comments and suggestions from the reviewers. The ϵ -Fe₂C catalysts prepared through our proposed method demonstrate unprecedented stability at temperatures around 440 °C (**Fig. 1a**), an achievement not reported in prior ϵ -Fe₂C research (ref. 21-24). This stability at 440 °C ensures that the ϵ -Fe₂C phase remains stable under typical FTS conditions. We think that comparing thermal stability with other state-of-the-art ϵ -Fe₂C catalysts (ref. 21-24) is not

particularly meaningful, and replicating these other catalysts perfectly for use as a reference in our study is challenging. The method of ϵ -Fe₂C obtained directly through the carburization of alpha-Fe are remarkably similar to those reported in *ref. 21 and 22*. Therefore, we believe this type of ϵ -Fe₂C adequately represents the typical properties of ϵ -Fe₂C without N, and the comparison underscores our hypothesis: The remarkable stability of our ϵ -Fe₂C derived from PBAs is likely due to nitrogen incorporation.

We extend our sincere gratitude to the reviewer for the invaluable suggestions aimed at enhancing the novelty of our work. Following your expert guidance, we have refined the presentation of our manuscript. (see title, abstract, Page2 line 54-58, Page4 line91-101, Page5 line128-137, Page6 line156-167, Page10 line253-273, Page11 line277-286) These efforts have significantly elevated the quality of our work. We have detailed the novelty of our work and the advancements over these Fe₂C systems (*ref 21 to 24*) in the response mentioned above, and will not discuss them in depth here. It should be noted that the superior selectivity for higher olefins and low C1 byproducts exhibited by the ϵ -Fe₂C-Mn system is not due to the N-induced catalyst preparation method, but rather due to the interfacial structure of highly dispersed MnO clusters and ϵ -Fe₂C.

We trust that this clarification will further illustrate the significant advancements our work contributes to the field and support the case for publication in *Nature Communications*.

3. The authors claimed that their catalysts showed suppressed C1 products in comparison to state-of-art literature, as illustrated in Figure 2C. However, the results of ref. 21 to 24 are missing from the comparison in Figure 2C, and those results showed comparable or better suppression of C1 products. Hence, the reviewer is not convinced by this claim for significant novelty.

Author reply: We sincerely appreciate the reviewer's valuable comments, which have prompted us to reflect more deeply on our work and have assisted us in presenting the novelty of our research more clearly. Regarding the missing comparative data from *ref. 21 to 24* that you mentioned, we have provided a detailed explanation and the measures taken following revision.

In the original version of **Fig. 2c**, our results primarily compared our catalysts with recent developments in FTO and Oxe-Zeo systems, as these studies were dedicated to enhancing olefin selectivity. Although these studies are significant in the field of ϵ -Fe₂C

(ref. 21 to 24), they did not focus on enhancing selectivity towards high-value olefins, which was a key goal of our study. Additionally, the data concerning olefin selectivity in these studies were either incomplete or not reported. Moreover, the catalysts tested in ref. 21 and 22 were evaluated under FTS conditions below 230°C, which significantly differ from the operational conditions of our study (280 – 300 °C), affecting their comparability. For these reasons, we chose not to include the comparative data from references 21 to 24 in Figure 2c.

Following your constructive suggestion, we have incorporated the results from *ref. 21 to 24* into **Supplementary Table 12 and added Fig. R6**, which demonstrates our work's significant improvements over the state-of-the-art Fe₂C catalysts. This addition emphasizes the unique performance advantages of our ε-Fe₂C-Mn system in suppressing C1 by-product selectivity. For instance, under low-temperature conditions (235 °C and 15% CO conversion), the ε-Fe₂C reported in ref. 22 shows a CO₂ selectivity of 5% and a methane selectivity of 17%. In contrast, even at a higher temperature of 280 °C, our ε-Fe₂C-Mn catalyst demonstrates only 2.4% CO₂ selectivity and 9.9% methane selectivity. These results not only highlight the enhanced efficiency of our Mn-modified ε-Fe₂C in minimizing C1 by-products but also affirm its superior performance in achieving high olefin selectivity compared to the established state-of-the-art ε-Fe₂C catalysts reported in *ref. 21 to 24*. We have added above discussion in the revised manuscript (see Page 10 line261-273) and included **Fig. R6 as Supplementary Fig.12**.

We are grateful for this opportunity to expand our dataset and refine our comparative analysis, and we believe these adjustments provide clear evidence of the novelty and significant advancements of our research.

Supplementary Table 12 | Comparison of the catalytic performance of ε-Fe₂C-Mn with syngas-to-olefins systems and other Fe₂C catalysts reported in the literature.

Entry	Catalysts	WHSV (ml·g _{cat} ⁻¹ ·h ⁻¹)	T (°C)	H ₂ /CO ratio	CO Conv [%]	Sel. [mol%]						Ref.
						CO ₂	CH ₄	C1 Total	C ₂₋₄	C ₅₊	Olefins	
1	ZnCrOx-MSAPO	5143	400	2.5	17.0	45.0	1.2	46.2	39.4	5.4	47.2 ^a	(1)
2	CoMnC/PDVB	1800	250		63.5	46.3	2.6	48.9	39.8	11.3	38.3 ^a	(2)
3	0.5Na/CoMnAl@6.6Si	4000	260	0.5	13.5	16.7	4.3	21.0	36.9	42.1	61.1	(3)
4	CoMn	2000	250	2	31.8	47.3	2.6	49.9	33.1	17.0	60.8 ^a	(4)
5	Co ₁ Mn ₃ -Na ₂ S	-	240	2	0.8	<3	17	<20	-	-	54.0	(5)
6	Na-FeCx/s-ZSM-5	2400	260	1	82.5	46.6	1.6	48.2	20.3	30.8	38.4 ^a	(6)
7	Fe/α-Al ₂ O ₃	1500	340	1	80.0	40	6.6	46.6	-	-	31.8 ^a	(7)
8	FeMn@Si-c	4000	320	2	56.1	13.0	10.0	23.0	-	-	64.9	(8)
9	Fe-Zn-0.81Na	60000	340	2.7	77.2	23.8	9.7	33.5	25.9	40.6	52.5	(9)

10	Fe ₃ O ₄ @MnO ₂	3000	280	1	67.9	47.1	3.6	50.7	-	-	41.9	(10)
11	ε-Fe ₂ C	18000	235	1.5	15.0	5.0	17.0	22.0	29.0	49.0	-	(11)
12	RQ-ε-Fe ₂ C	-	170	2	76.0	19.0	18.0	37.0	28.4	34.6	-	(12)
13	ε-Fe ₂ C@graphene	-	300	1	-	20.3	8.2	28.5	19.0	52.4	13.4 ^a	(13)
14	ε-Fe ₂ C/Al ₂ O ₃	9000	280	1	-	19.2	9.0	28.2	25.3	46.5	18.0 ^a	(14)
15	ε-Fe ₂ C-Mn	20000	300	2.5	41.0	11.9	7.1	19.0	42.8	38.2	70.2	This work
16	ε-Fe ₂ C-Mn	60000	280	2.5	17.1	2.4	9.9	12.3	52.8	34.9	71.3	This work

a: The values denote the selectivity and yield of lower olefins (C₂₋₄ =)

4. This study delved deeply into the impact of Mn, a widely recognized promoter. The results obtained align closely with existing literature findings.

Author reply: Thank you for the reviewer's comments. We appreciate the opportunity to clarify the innovative aspects of our work, particularly concerning the use of Mn as a promoter in iron-based FTS.

While Mn is indeed a well-researched promoter known to inhibit hydrogenation reactions, decrease methane selectivity, and increase olefin selectivity—as noted in the introduction of our manuscript—our research goes beyond reaffirming these established effects. We are the first to report the preparation of Mn-modified ε-Fe₂C catalysts through the pyrolysis of PBAs. This novel approach enables high olefin production from syngas while simultaneously suppressing C1 byproducts, as demonstrated in our study.

Our results not only align with but also significantly advance past findings by showing that our Fe₂C-Mn system achieves a CO₂ selectivity of only 11.9% and a total C1 by-product selectivity of less than 20%. This is a notable improvement over the 22% C1 by-product selectivity reported by Ding *et al.* (*Science* **2021**, 371, 610) using a hydrophobic catalyst. Moreover, a comparative analysis with extant literature on Mn promoters (refer to **Table R3**) reveals that most studies report a selectivity for C1 by-products exceeding 35%. This marked improvement in C1 selectivity is indicative of enhanced carbon utilization efficiency. To our knowledge, existing literature on the Fe-Mn system consistently reports that Mn promoters enhance olefin selectivity, but there are no reports of a significant reduction in CO₂ selectivity.

Furthermore, to address the unique contributions of our work: We have demonstrated both experimentally and theoretically, how the interfacial structure formed by highly dispersed MnO clusters and ε-Fe₂C inhibits the conversion pathway from CO to CO₂ (**Fig. 4**). This finding is not only a validation of the Mn promoter's role but also an advancement in understanding its mechanistic impact on CO₂ selectivity

reduction. We have added above descriptions in revised manuscript (see Page10, line253-260) and included **Table R3** as **Supplementary Table 11**.

We trust that this explanation underscores the significant advancements our study contributes to the existing body of knowledge and illustrates that while our findings corroborate the known effects of Mn, our innovative methods and insights provide unprecedented advancements in the field. We hope this clarification satisfactorily addresses the concerns raised and supports the significance and originality of our work.

Table R3 Comparison of the performance of ϵ -Fe₂C-Mn and Mn-promoted Fe-based catalysts reported in the literature.

Entry	Catalysts	WHSV (ml·g _{cat} ⁻¹ ·h ⁻¹)	T (°C)	H ₂ /CO	CO Conv [%]	Sel. [mol%]						Ref.
						CO ₂	CH ₄	Cl Total	C ₂₋₄	C ₅₊	Olefins	
1	Mn/ γ -Fe ₂ O ₃	4480	320	1	57.1	31.7	8.0	39.7	47.9	12.4	41.8 ^a	Appl. Catal. B 2020, 261, 118219
2	Fe-MnK-AC	3000	320	1	85.0	48	11.8	59.8	24.8	15.4	20.5 ^a	Appl. Catal. A 2017, 541, 50.
3	FeMnLi	5000	320	2	85.6	34.6	9.3	43.9	32.5	23.6	24.0 ^a	Fuel 2019, 257, 116101
4	Fe ₃ O ₄ @MnO ₂	11000	340	2	91.8	37.9	7.5	45.4	26.2	28.4	23.2 ^a	Ind. Eng. Chem. Res. 2019, 58, 21350.
5	FeMnCu	1500	300	2	96.9	23.0	15.4	38.4	54.3	7.3	30.9 ^a	Appl. Catal. B 2020, 278, 120683
6	Fe-Mn(4:1)	1500	260	1	5.5	20.7	15.0	35.7	53.8	10.5	38.6 ^a	Appl. Catal. B 2021, 285, 119815
7	Mn/Fe ₃ O ₄	4480	320	1	41.5	37.8	6.0	43.8	41.5	14.7	37.4 ^a	ACS Catal. 2015, 5, 3905.
8	MnxFe _{3-x} O ₄	4000	260	1	7.1	25.2	12.7	37.9	52.1	10.0	40.7 ^a	J. Catal. 2020, 381, 150.
9	Fe ₄ Mn ₁	7500	280	1	32.4	42.8	11.3	54.1	37.3	8.6	29.3 ^a	J. Catal. 2023, 417, 213.
10	100Fe7Mn	2000	250	2	45.1	19.2	9.6	28.8	35.3	35.9	26.8 ^a	J. Energy Chem. 2013, 22, 624
11	Mn-KCuFe/mAl ₂ O ₃	2000	270	1.25	95.0	39.4	6.1	45.5	5.5	49.0	4.7 ^a	Appl. Catal. A: Gen. 2020, 607, 117861
12	FeMnCu/MCF-0	6000	270	1	40.9	11.7	18.4	30.1	44.8	25.1	28.8 ^a	Catal. Sci. Technol. 2020, 10, 502
13	Fe@12.42Mn	4000	265	2	42.0	14.4	8.4	22.8	36.2	41.0	28.7 ^a	Fuel 2024, 360, 130567
14	Fe _{2.86} Mn _{0.14} O ₄ /CNT	6000	300	1	43.9	37.2	3.8	41.0	25.2	33.8	19.8 ^a	Catal. Today 2013, 215, 86
15	ϵ -Fe ₂ C-0.4Mn	20000	300	2.5	41.0	11.9	7.1	19.0	42.8	38.2	70.2	This work
16	ϵ -Fe ₂ C-0.4Mn	60000	280	2.5	17.1	2.4	9.9	12.3	52.8	34.9	71.3	This work

a: The values denote the selectivity and yield of lower olefins (C₂₋₄)

REVIEWERS' COMMENTS

Reviewer #2 (Remarks to the Author):

I am satisfied with the authors' revisions to my concerns and recommend the acceptance of the revised manuscript in Nat. Comm.

Reviewer #3 (Remarks to the Author):

I appreciate the efforts and thoughts that the authors put in to the revision, and congratulates the authors on this excellent piece of work. My feedback has been fully addressed by the authors so this revised version is recommended for publication in Nature Communication.

REVIEWERS' COMMENTS

In this response letter, the reviewers' comments are presented in *black italics*, our responses are in blue.

Reviewer #2 (Remarks to the Author):

I am satisfied with the authors' revisions to my concerns and recommend the acceptance of the revised manuscript in Nat. Comm.

Author reply: Thank you for the reviewer's positive feedback regarding the revisions made to address your concerns. We sincerely appreciate your thorough review and recommendation for the acceptance of our revised manuscript in Nature Communications.

Reviewer #3 (Remarks to the Author):

I appreciate the efforts and thoughts that the authors put in to the revision, and congratulates the authors on this excellent piece of work. My feedback has been fully addressed by the authors so this revised version is recommended for publication in Nature Communication.

Author reply: We would like to express our sincere gratitude for your kind words and for recognizing the efforts and thoughtfulness we have put into revising our manuscript. The reviewer's feedback has been invaluable in guiding us towards improving the quality of our work, and we are truly honored by the reviewer's commendation of our efforts. We are thrilled to hear that the reviewer found the revised version of our manuscript to be satisfactory and that all concerns have been adequately addressed. It is truly encouraging to receive such positive feedback from the reviewer, and we are grateful for the reviewer's recommendation for publication in Nature Communications.